# Zebrafish capable of generating future state prediction error show improved active avoidance behavior in virtual reality

Makio Torigoe [1], Tanvir Islam[1,2], Hisaya Kakinuma[1,2], Chi Chung Alan Fung[3], Takuya Isomura [4], Hideaki Shimazaki [5], Tazu Aoki[1], Tomoki Fukai [3] & Hitoshi Okamoto [1,2 ✉]

Animals make decisions under the principle of reward value maximization and surprise minimization. It is still unclear how these principles are represented in the brain and are reflected in behavior. We addressed this question using a closed-loop virtual reality system to train adult zebrafish for active avoidance. Analysis of the neural activity of the dorsal pallium during training revealed neural ensembles assigning rules to the colors of the surrounding walls. Additionally, one third of fish generated another ensemble that becomes activated only when the real perceived scenery shows discrepancy from the predicted favorable scenery. The fish with the latter ensemble escape more efficiently than the fish with the former ensembles alone, even though both fish have successfully learned to escape, consistent with the hypothesis that the latter ensemble guides zebrafish to take action to minimize this prediction error. Our results suggest that zebrafish can use both principles of goal-directed behavior, but with different behavioral consequences depending on the repertoire of the adopted principles.

[1] Lab. for Neural Circuit Dynamics of Decision Making, RIKEN Center for Brain Science, Wako, Saitama 351-0198, Japan. [2] RIKEN CBS-Kao Collaboration Center, Wako, Saitama 351-0198, Japan. [3] Neural Coding and Brain Computing Unit, Okinawa Institute of Science and Technology, Onna-son, Okinawa 904-0495, Japan. [4] Brain Intelligence Theory Unit, RIKEN Center for Brain Science, Wako, Saitama 351-0198, Japan. [5] Center for Human Nature, Artificial Intelligence, and Neuroscience (CHAIN), Hokkaido University, Sapporo, Hokkaido 060-0812, Japan. ✉email: hitoshi.okamoto@riken.jp

Making optimal decisions according to the current sensory input is essential for animals. One prevailing model underlying this behavioral process is based on the idea that the ultimate aim of choice is to maximize utility or reward[1]. In addition to this, adaptive behavior requires animals to generate an internal model of their environment and to take actions to minimize surprise (i.e., improbability) about the state they encounter in comparison with the state predicted from the internal model[2,3]. How these mechanisms are actually adopted by animals and are reflected in their behavior remains unknown[4,5].

Active avoidance has been regarded as the most typical model-free decision-making behavior based solely on the basic principle of reinforcement learning. The goal of reinforcement learning is to maximize the predicted reward under a given spatial distribution of utility (reward or punishment associated with spatial subregions)[6–9]. Another control process of this goal-directed behavior could underlie the principle of minimization of surprise (i.e., the prediction error between the real perceived state and the predicted state) and has been theoretically formulated as "active inference"[2,3]. However, there has been little experimental data to confirm whether active inference indeed acts to correct behaviors.

Adult zebrafish have the ability to learn various adaptive behaviors, and their telencephalon has regions and neural circuits that are evolutionarily homologous to those of other vertebrates, including mammals[10]. These regions include the isocortex, hippocampus, amygdala, and the cortico-basal ganglia circuit, which is implicated in behavioral selection[11–14]. The zebrafish brain is very small ($3 mm^3$)[15] as compared to that of mice ($509 mm^3$)[16] or humans ($1400 cm^3$)[17], allowing us to observe neural activity in a relatively wide brain region[8]. Further, the use of pigment-deficient mutant strains[18,19] enables observation of the telencephalic neural activity without opening the skull. Thus, adult zebrafish can be an attractive animal model for investigating the evolutionarily conserved and universal mechanisms of behavioral control by the telencephalon.

To address whether zebrafish are able to use both value maximization and surprise minimization[2,3], we established a closed-loop virtual reality two-photon calcium imaging system in which the surrounding scenery moved backward in response to the tail beating of the fish (Fig. 1a). We used this system to study active or passive avoidance behavior as an example of the simple behavioral paradigm of goal-directed behaviors[6–9]. Our unsupervised analysis method, non-negative matrix factorization (NMF)[20,21], partitioned the complex activity pattern of the entire neural population into a linear superimposition of the neural activity of multiple elemental ensembles.

Using this virtual reality system, we found that adult zebrafish can learn both GO (active avoidance) and NOGO (passive avoidance) tasks in a series of trials carried out within one day. These tasks are commonly used also in mice and monkeys[22–25]. Furthermore, by taking advantage of the virtual reality environment, we could change the task conditions at will or even impose conditions that would be impossible in the real world, e.g., converting the system from a closed-loop to an open-loop condition, where the scenery did not respond to the fish tail beating.

Our results suggest that some zebrafish, if not all, can use both of the two principles for achieving safe escape in active avoidance, i.e., value maximization and surprise minimization. Further, under limited time constraint of the trials, zebrafish using both principles exhibit optimized escape behavior compared with zebrafish behaving with only the reward-based principle.

Our work enables to study the cellular and molecular mechanisms of action selection in vertebrates in the context of the entire neural circuits at the finer levels that are impossible to achieve using higher vertebrates.

## Results

### Establishment of the closed-loop virtual reality system with two-photon real-time imaging of telencephalic neural activity in transparent adult zebrafish.
To reveal the mechanism of appropriate behavior selection, we established a closed-loop virtual reality two-photon calcium imaging system (Fig. 1a). The calcium signals captured by this system, the top view of the telencephalic region in tethered fish and the actual tail behavior and movement of feedback scenery are shown in Supplementary Movies 1, 2, and 3, respectively.

We fixed the heads of adult zebrafish by attaching a custom-made harness with dental bond and cement (Fig. 1b). The detailed procedure and apparatus are shown in Supplementary Fig. 1a, b. This method enabled continuous imaging of the beating tail and the neural activity of a wide area of the adult zebrafish telencephalon during the trials. The head-fixed fish were put in a small tank surrounded by four liquid-crystal displays (LCDs) on the left, right, front, and bottom of the tank (Fig. 1a). The system alternated between the presentation of the visual stimuli on the four LCDs and the imaging of the neural activity by the photomultiplier tube (PMT) of the microscope (Fig. 1e). During the scanning of each line (650.24 μs), the gallium arsenide phosphide (GaAsP)-PMT detector (Zeiss, BIG detector) was switched on and the displays were switched off (Fig. 1e, green arrows). During the shift to the next line (~130 μs), GaAsP-PMT was switched off and the displays were switched on (Fig. 1e, blue arrows). To achieve this, we selected the TTL signal that was generated at the onset of each line scanning from the two-photon microscope. The electronic stimulator (SEN-3401, NIHON KOHDEN) received this TTL signal and sent a modified time TTL signal based on the original TTL signal derived from the microscope. This TTL signal from the stimulator was used to switch the display on and off. The GaAsP-PMT switch was controlled by the custom-made system by Zeiss.

As the fish beat their tail, the visual images of the scenery presented on all the surrounding displays on the left, right, front, and bottom sides of the tank were shifted backward according to the calculated virtual traveling distance of the fish to make the system closed-loop (Fig. 1a and "Methods"). The actual tail behavior and the feedback scenery movement are shown in Supplementary Movie 3. We performed calcium imaging of the neural activity in a wide surface area of the telencephalon ($384.9 × 384.9$ μm) using a piezo actuator, which allowed us to capture three or six slices of images separated by 16 μm (approximately 300 ms for three-slice imaging, and thus ~3 Hz: approximately 600 ms for six-slice imaging, and thus ~1.5 Hz; Fig. 1c, d).

The zebrafish telencephalon is reported to contain the homologous regions of the isocortex (central zone of the dorsal pallium, Dc), hippocampus (lateral zone of the dorsal pallium, Dl), and amygdala (medial zone of the dorsal pallium, Dm[26]; Fig. 1c). The best way to reveal the role of these regions in decision-making would be to capture neural activity from all of these regions. However, due to various technical limitations of conventional two-photon calcium imaging at present, we focused our observation of neural activity imaging mainly on the surface part of Dc, the putative zebrafish homolog of the mammalian isocortex, because the isocortex plays an important role in mammalian decision-making. (Fig. 1c, d). The area we observed occupied the large surface part of the Dc and a part of the other dorsal pallial regions, such as Dm and Dl (Fig. 1c, d). To achieve the extensive labeling of the excitatory neurons, we used a triple transgenic line for *camk2a: gal4, vglut2a: gal4*, and *UAS: G-CaMP7* (Fig. 1d).

### Zebrafish can learn to avoid shock by training in virtual reality.
To reveal the mechanism of adaptive goal-directed behavior, we designed GO and NOGO tasks in the virtual reality system

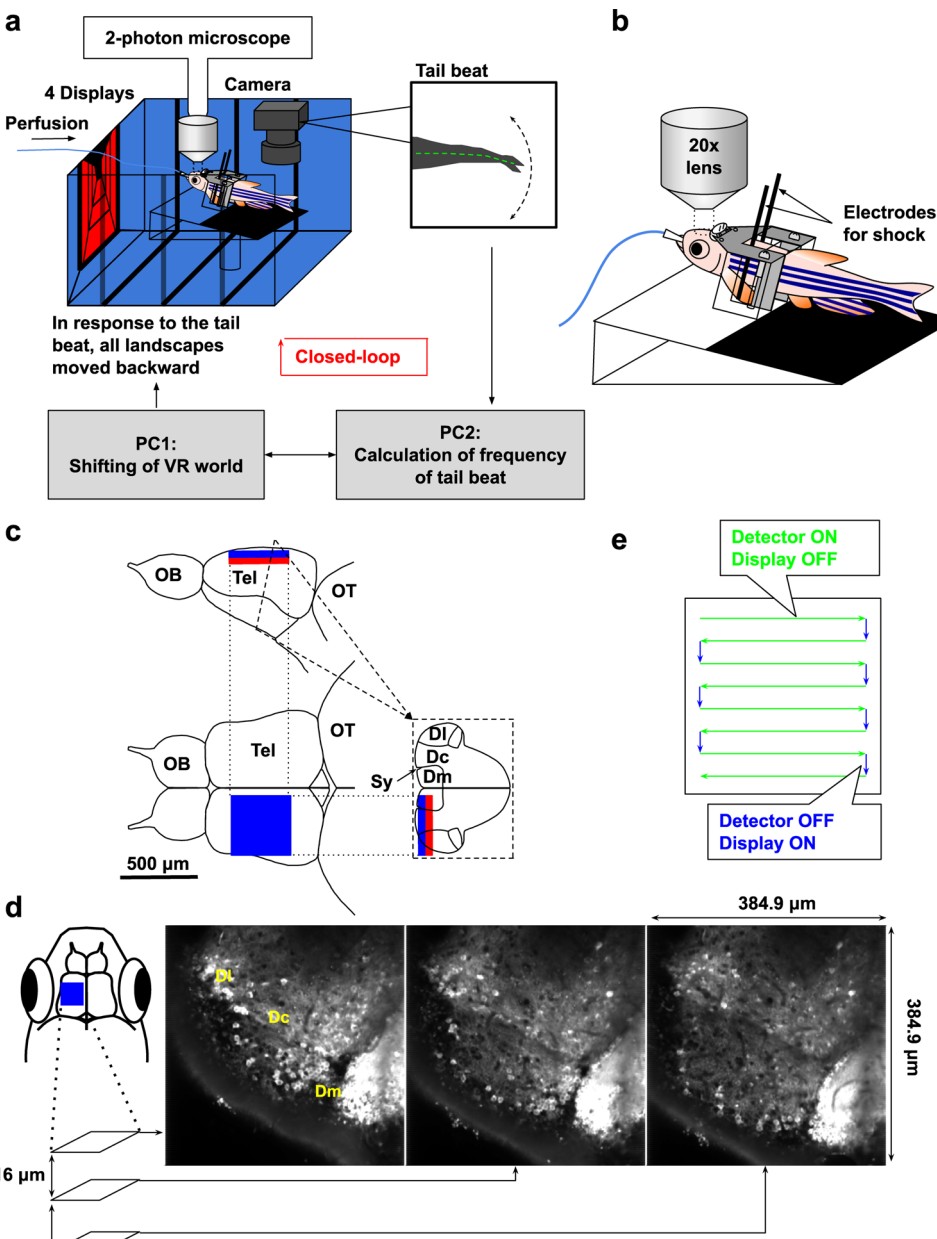

**Fig. 1 The closed-loop virtual reality two-photon imaging system enables real-time capturing of neural activity in adult zebrafish. a** Schematic diagram of the closed-loop virtual reality setup. Four displays presented visual stimuli. Tail beating was captured by a camera and caused the scenery to move backward to create the impression of forward swimming. The virtual traveling distance was calculated by [frequency of tail beats] × [gain]. **b** Schematic drawing of the tethered adult zebrafish using a custom-made harness, dental bond, and cement. Two needle electrodes were placed on both sides of the body to deliver electric shocks. **c** The imaged region in the telencephalon. The side (top) and dorsal (bottom left) views and coronal section (bottom right) of the adult zebrafish brain. The blue box indicates the imaged region by surface three-plane imaging, and the red box indicates the additionally imaged region by six-plane imaging. Dc, central zone of dorsal telencephalic area; Dl, lateral zone of dorsal telencephalic area; Dm, medial zone of dorsal telencephalic area; OB, olfactory bulb; OT, optic tectum; Sy, sulcus ypsiloniformis; Tel, telencephalon. **d** Calcium imaging of neural activity in three focal planes using the piezo actuator. Either left or right hemisphere was imaged. These images are averaged images of the left hemisphere in three focal planes. Anterior to top; lateral to left; medial to right. Dl, lateral zone of the dorsal telencephalon; Dc, central zone of dorsal telencephalon; Dm, medial zone of dorsal telencephalon[10]. **e** Schema of alternate switching of neural activity detection by a two-photon microscope and visual stimulation by displays. Green arrows indicate the duration of scanning; in this setting, the detector is ON and displays are OFF. Blue arrows indicate the duration from the end of a line scan to the onset of the next line scan; in this setting, the detector is OFF and displays are ON.

(Fig. 2a). After the inter-trial interval (15 s) when fish perceived white color with black stripes on the surrounding areas of the displays, the GO or NOGO task was randomly initiated. In the GO task, the surrounding color turned blue, and fish had to escape to the red region in front of them within 10 s. For the NOGO task, the surrounding color turned red, and the fish had to stay in the red region for 10 s. If the fish did not behave as indicated by the given task conditions, an electric shock (5 V/cm for 1 s) was delivered from two needle electrodes on both sides of the body (Fig. 1b). The fish had to determine whether to move forward or to stay depending on the presented color to avoid electric shock.

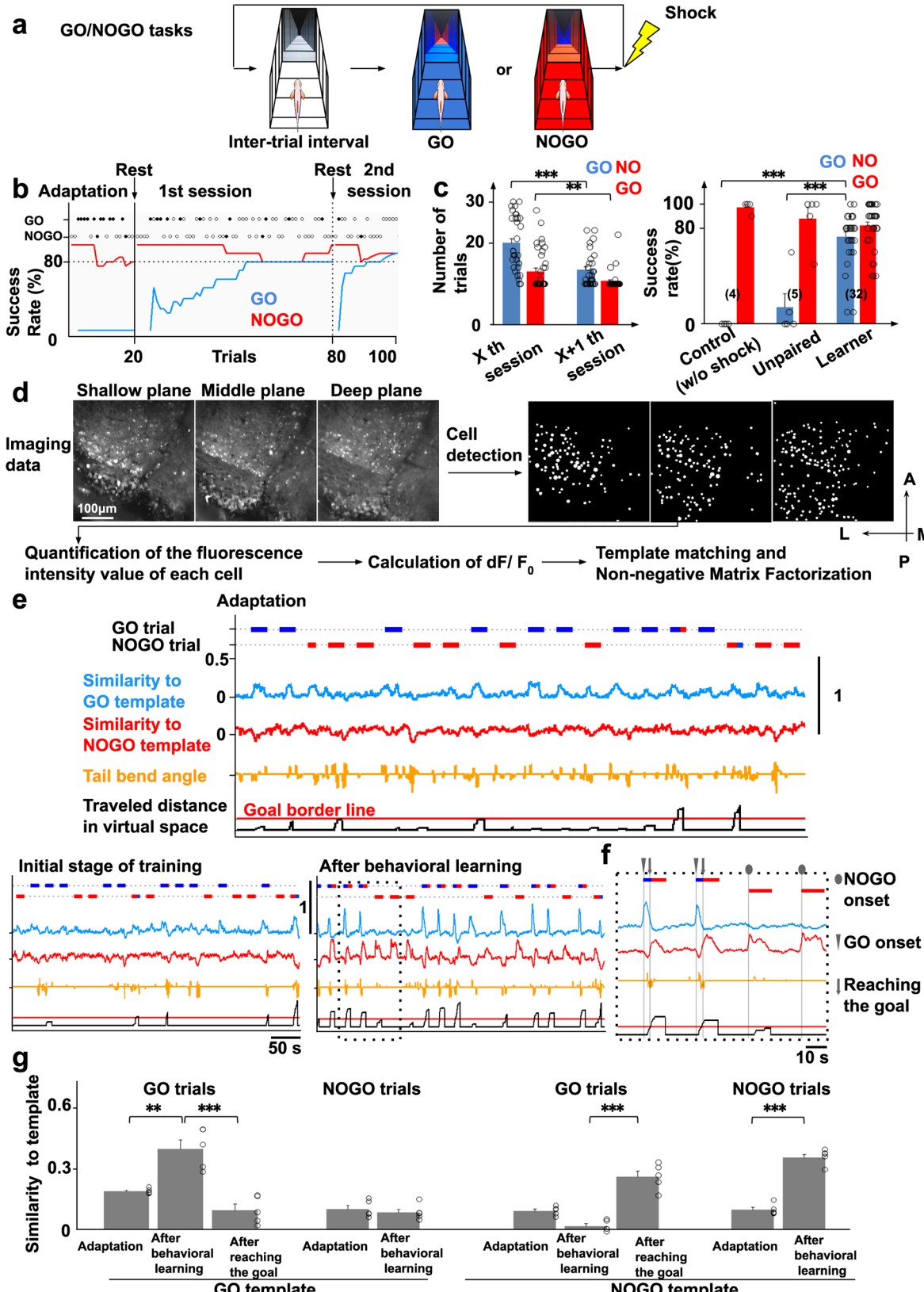

Figure 2b presents the learning curve of the GO/NOGO trials. During the adaptation stage in which fish perceived color changes as shown in Fig. 2a but electric shock was not provided, the fish tended to stop. Therefore, the apparent success rate of the NOGO trials tended to be high from the beginning of the session. In contrast, the success rate of the GO trials was low at the beginning. However, as training proceeded, the fish gradually succeeded in the GO trials and eventually, the success rate of both GO and NOGO trials met the learning criterion, i.e., 80%, in both trials, calculated from the past ten trials. We regarded that the fish had learned at this time point. In the second session, the success rate of both GO and NOGO trials met the learning criterion

**Fig. 2 Fish can learn the GO/NOGO tasks in the virtual reality system and the specific neural population of the telencephalon is activated when fish perceive blue color. a** GO/NOGO tasks in the virtual reality system. **b** The learning curve of GO/NOGO trials of a fish that met the behavioral learning criterion. Horizontal dotted line, the criterion for behavioral learning; open circles, successful trials; solid circles, failed trials; vertical line, the initiation time point of trials with electric shock; vertical dotted line, the initiation of the next session. **c** Left: the number of trials needed until the behavioral learning criterion of GO and NOGO trials was satisfied in the Xth session and the next X + 1th session (32 fish). Xth GO vs X + 1th GO, [***]$P = 7.72 \times 10^{-6}$; Xth NOGO vs X + 1th NOGO, [**]$P = 4.03 \times 10^{-3}$. Two-tailed paired t-test. Right: Comparison of the success rates among control, unpaired, and learner groups at the 22nd GO trial and16th NOGO trial which were the average numbers needed to achieve the behavior criteria in standard learner fish. Control (w/o shock) vs learner, [***]$P = 7.52 \times 10^{-8}$; unpaired vs learner, [***]$P = 2.27 \times 10^{-6}$, two-tailed unpaired t-test. Columns and error bars: mean ± SEM. Each circle represents one fish. The numbers in parentheses are the number of fish used in the statistics. **d** Imaging data analysis procedure. See "Methods" for details. **e** Similarity to GO and NOGO templates in the adaptation stage (upper panel), initial stage of training (bottom left panel), and after the establishment of behavioral learning (bottom right panel). Horizontal bars in the upper and lower positions indicate the period of GO and NOGO trials, respectively. The bar colors indicate the color of the environment at the position of the fish. **f** Enlarged view of the boxed area in (**e**, bottom right panel). **g** Comparison of peak value of the similarities to both GO and NOGO templates in both trials in the adaptation stage, after behavioral learning, and after reaching the goal after behavioral learning. Columns and error bars: mean ± SEM. Circles in the adaptation stage and after behavioral learning indicate peak similarities when the fish was in the start color in the first five trials. Circles after reaching the goal indicate peak similarities after fish reached the goal in the first five GO trials after behavioral learning. Adaptation vs after behavioral learning in GO trials GO template, [**]$P = 1.56 \times 10^{-3}$; after behavioral learning vs after reaching the goal in GO trials GO template, [***]$P = 4.5 \times 10^{-4}$; after behavioral learning vs after reaching the goal in GO trials NOGO template, [***]$P = 5.02 \times 10^{-5}$; adaptation vs after behavioral learning in NOGO trials NOGO template, [***]$P = 1.56 \times 10^{-6}$. Two-tailed unpaired t-test.

again. Of 129 fish, 45 met the learning criterion, of which 33 satisfied the learning criterion again in the following session. Of these 33 fish, the imaging data of 32 exhibited little z-axis deviation during the imaging period and could be used for analysis. In these 32 fish, the number of trials required to meet the learning criterion again after first reaching it decreased (Fig. 2c, left panel). Compared with the control group (without shock, four fish) and the unpaired group (with shock in the inter-trial interval, five fish), the rate of successfully swimming forward in the GO trials was higher in the learner group (Fig. 2c, right panel), suggesting that fish could learn to escape by beating the tail when perceiving blue color in the GO trials and could retain the learned response. In contrast, because of the fish's tendency to stop tail beating under the tethered condition in the VR arena, we could not conclude that fish actually learned to stay in the red region in NOGO task only from these behavioral data. However, we detected certain change in the neural activities following repeated NOGO trials as described below.

**Template-matching analysis revealed that a similar neural population was activated when fish perceived blue color in the GO trials after behavioral learning was established.** To reveal the neural basis of the appropriate behavior selection, we analyzed the calcium imaging data of the telencephalic neural activity. We first identified regions of interest (ROIs) corresponding to cells and calculated the fluorescent change ($\Delta F/F_0$) in each cell (Fig. 2d). After correcting the x–y axis displacement of the images obtained in the experiment, the ROIs corresponding to the cells were defined based on a previously published method[27]. In more detail, we calculated the "peaky-ness" corresponding to the time change in fluorescence intensity for all pixels. Focusing on the pixel with the highest peaky-ness, we calculated the correlation between the temporal change in the fluorescence intensity of the focused pixel and that of the surrounding pixels, and the pixels with a correlation above threshold were regarded as one cell. We then focused on the pixel with the second-highest peaky-ness and performed the same calculation as before. This process was continued until the peaky-ness was lower than the set threshold. After detection of all cells, the cell with the largest spatial size was used if there was a spatial overlap between cells.

To determine how the recruited neural population changed with behavioral learning, we first performed a template-matching analysis[28]. For this analysis, we created a template by averaging individual neurons' activity from 0 to 2 s after the onset of the

trial in the successful GO trials or the successful NOGO trials after fish reached the learning criteria. We calculated the similarity with the template (Fig. 2e) at each time point by sliding the template from the beginning to the end of the session. The value of the similarity index varied from −1 (anticorrelation) to +1 (perfect correlation). Of this index, 0 means no correlation. When we calculated with the GO trial template, the similarity at the onset of the GO trial increased after behavioral learning criteria was met, but not in the adaptation stage in 27 of the 32 fish (Fig. 2e, g, circles indicate the peak values in each trial, and see also filled pentagonal stars of each fish in Supplementary Fig. 2 where the similarity indices at various stages of training are shown for all 32 examined fish. The numbers attached to each graph in Supplementary Fig. 2 correspond to the fish numbers in Supplementary Table 1.). We obtained similar results when using the NOGO template in 22 of the 32 fish. Again, the similarity to the template was greater at the beginning of the NOGO trial after the behavioral learning criteria were met, but not in the adaptation and initial stages of training (Fig. 2e, g, Supplementary Fig. 2 (filled hexagonal star in each fish), and Supplementary Table 1). When we conducted the analysis with the NOGO template, in six of the 22 fish, the similarity increased also after reaching the red goal in the successful GO trials after the behavioral learning criteria were met (Fig. 2f, g, Supplementary Fig. 2 (filled heptagonal star), and Supplementary Table 1). These results suggest that a similar neural population was activated each time the fish perceived blue color in the GO trial after the behavioral learning criteria were met. Further, although the behavioral data indicated no difference in the success rate of NOGO trials between the early and late stage of training (Fig. 2c, right panel), the template-matching analysis revealed that similar neural populations started to be activated when a fish perceived red color, both at the onset of the successful NOGO trials and after reaching the safe red goal in the successful GO trials at a late stage of training when behavioral learning for GO trials was established. The similarity increase was not observed in the adaptation and initial stages of training.

**Non-negative matrix factorization analysis revealed multiple neural ensembles with one encoding the perception of blue color.** To further reveal the nature of the specific neural populations, which are impossible or difficult to identify by template matching or by any other methods, such as regression analysis or principal component analysis, we performed an unsupervised analytical method, NMF (Fig. 3a)[20,21]. The timeline of neural

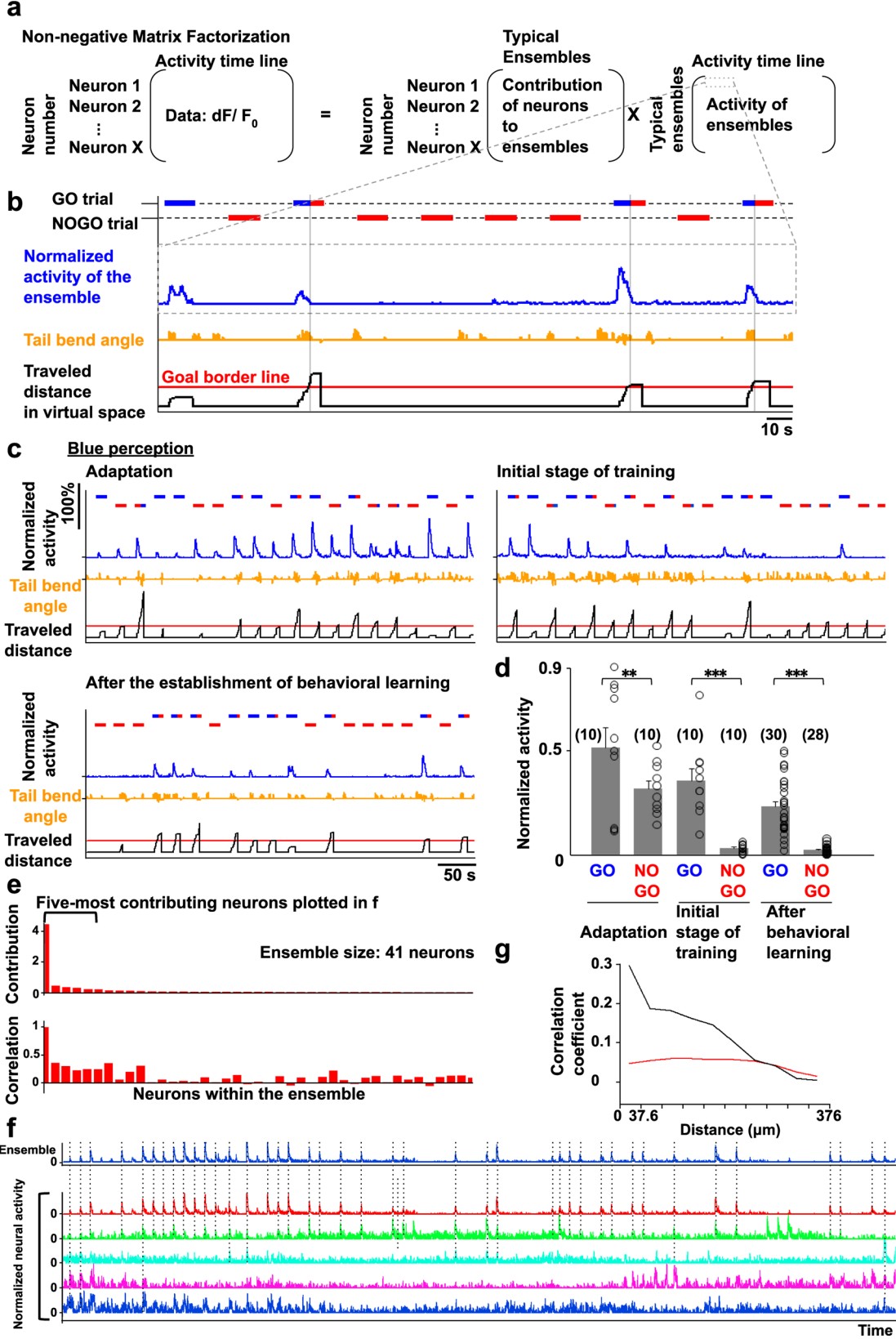

activity can be expressed by one matrix with each row corresponding to the time-lapse activity of each neuron. NMF factorizes this matrix into two, in which the columns of the first matrix present the levels of contribution of individual constituent neurons in typical ensembles of neural activity and the rows of

the second matrix indicate how frequently each ensemble is activated at each time point. The number of ensembles was determined according to the Akaike Information Criteria (AIC). The AIC curves of all learner fish are shown in Supplementary Fig. 3. We compared the activity timeline with the behavioral data

**Fig. 3 Non-negative matrix factorization analysis revealed the ensemble encoding the perception of blue color. a** The formula to calculate the non-negative matrix factorization (NMF; for details, see "Methods"). **b** The activity of the blue perception-coding ensemble as an example of an activity pattern (the activity in this panel is a part of **c**, bottom left panel). The blue line indicates the activity of a neural ensemble normalized by the self-maximum value. The horizontal bars in the upper and lower positions indicate the period of GO and NOGO trials, respectively. The bar colors indicate the color of the environment at the position of the fish. Orange line, the tail bend angle. Black line, the distance that fish had traveled. Red line, the point of color change. **c** The activity of the blue perception-coding ensemble in the adaptation stage (upper left panel), initial stage of training (upper right panel), and after behavioral learning was established (bottom left panel). **d** Quantified activity of the blue perception-coding ensemble in the adaptation stage, initial stage of training, and after behavioral learning was established. Columns and error bars: mean ± SEM. Circles indicate the peak value in each GO or NOGO trial. The numbers in parentheses are the number of trials used in the statistics. Adaptation GO vs adaptation NOGO, $^{**}P = 3.06 \times 10^{-3}$; initial stage of training GO vs initial stage of training NOGO, $^{***}P = 1.67 \times 10^{-5}$; after behavioral learning GO vs after behavioral learning NOGO, $^{***}P = 1.96 \times 10^{-6}$, $F(5, 92) = 26.25$. One-way ANOVA, Bonferroni's multiple comparison test. **e** Contribution of each neuron within the ensemble (upper panel) and correlation coefficient of each neuron's activity to the ensemble's activity (lower panel). **f** The activity of the ensemble (top trace) and the five most-contributing neurons in the ensemble (descending order from the top). Dotted lines indicate the timing when the neurons showed simultaneous activation with the ensemble. **g** Relationship between the correlation coefficient and distance for the 10 most-contributing neurons in the ensemble encoding the perception of blue. The data were averaged from 27 fish with this ensemble. Black line denotes the averaged correlation from 27 fish. Red line denotes the average of averaged 10 shuffled data from 27 fish (see "Methods").

to decipher the information encoded by each ensemble (see "Methods").

Consistent with the results of template-matching analysis, some of the neural ensembles obtained by the NMF analysis were always activated in the successful GO trials after establishment of the behavioral learning (see Supplementary Fig. 4, which representatively depicts the activities of all ensembles in one fish). Among them, a neural ensemble that was activated when the fish perceived blue color exhibited increased activity regardless of whether the fish had learned the behavior (Fig. 3b, c, blue line, 3d). All data concerning this fish (Fish 1 in Supplementary Table 1) are summarized in Supplementary Fig. 5 (see also Supplementary Figs. 6 and 7, and Supplementary Table 1 for other two fish, Fish 2 and Fish 3). Supplementary Figs. 5–7a show the activity changes of ensembles during training. This result suggests that this neural ensemble encodes the perception of blue color. The data obtained from Fish 1 (Supplementary Fig. 5) were used to generate Figs. 4e–h and 6g–i.

In this ensemble, approximately 20% of the total imaged neurons showed larger than zero contribution within the ensemble (Supplementary Fig. 8). To further reveal details of this ensemble encoding the perception of blue color, we plotted the contribution of each neuron within the ensemble (Fig. 3e, upper panel and Supplementary Figs. 5–7b, middle panel in left column, blue perception) and the correlation coefficient of each neuron's activity to the ensemble's activity (Fig. 3e, bottom panel and Supplementary Figs. 5–7b, bottom panel in left column, blue perception). Supplementary Figs. 5–7b show the spatial distributions of various neural ensembles (Supplementary Figs. 5–7b, upper panels), the contribution of neurons within the ensemble (Supplementary Figs. 5–7b, middle panels) and the correlation coefficient of each neuron's activity to the activity (Supplementary Figs. 5–7b, bottom panels) of each ensemble. Although one neuron had by far the highest level of contribution, other neurons with relatively lower contributions also showed significantly high levels of positive correlation. A further comparison between the ensemble activity and each neuron's activity revealed that the neurons with lower levels of contribution but with relatively high correlation coefficients showed coincidental activation with the ensemble, but in an intermittent manner (Fig. 3f and Supplementary Figs. 5–7c, upper left panels, blue perception). Supplementary Figs. 5–7c show the activities of the ensemble (top trace) and the five most-contributing neurons in the ensemble (descending order from the top).

To evaluate the distribution of neurons within the ensemble, we plotted the neurons with their weight of contribution in the ensemble, but we did not identify the brain region where the neurons within the blue perception ensemble preferentially accumulated. Supplementary Fig. 9 shows the distribution of the neurons within the blue perception ensemble across different fish.

However, to further evaluate the spatial distribution of the neurons within the ensemble, we calculated the correlation coefficient of the 10 neurons with the highest levels of contribution to the ensemble. We then plotted the relationship between the paired correlation coefficients and the distance of these neurons after averaging the data derived from all 27 fish that had the ensemble encoding blue color perception (Fig. 3g; for individual fish, please see Supplementary Figs. 5–7d, upper left panels, blue perception). The result showed that neurons with highly correlated activity have a tendency to be localized at a closer distance, implying that neurons in the zebrafish telencephalon encode the information in locally clustered populations. Supplementary Figs. 5–7d show the correlation between the correlation coefficients of paired neurons' activities and their distance in each ensemble for individual fish.

**Neural ensembles encoding the rules assigned to the colors blue and red.** Unlike the neural ensemble that simply encoded the perception of blue color, in 24 of the 32 fish (Supplementary Table 1, (+) in the "blue is dangerous" column), another neural ensemble showed increased activity when fish perceived blue color, but emerged only in repeated trials. This ensemble did not increase its activity during the adaptation and initial stages of training as shown with the cyan line in Fig. 4a (see also Fig. 4c; all data concerning Fish 2 in Supplementary Table 1 is summarized in Supplementary Fig. 6). This ensemble began to increase in activity before meeting the behavioral learning criterion (Fig. 4a, middle panel, cyan line, 4c, and Supplementary Figs. 5–7a, cyan line, blue is dangerous), and continued to display activation after behavioral learning was established (Fig. 4a, right panel, cyan line, 4c, and Supplementary Figs. 5–7a, cyan line, blue is dangerous). These data suggest that the activity of this ensemble did not merely represent the perception of blue color. The increased activity of this ensemble was also observed in the failed GO trials where fish did not perform the appropriate forward moving behavior (Fig. 4a, asterisks, cyan line and Supplementary Figs. 5–7a, cyan line, blue is dangerous). Focusing on one trial, the activity of this ensemble increased when the fish was presented with blue color and decreased as the fish reached the goal (Fig. 4b, cyan line). These results suggest that the increased activity of this ensemble was observed when fish perceived the blue color paired with shock in repeated trials.

In 20 of the 32 fish (Supplementary Table 1, (+) in the "red is safe" column), we observed another ensemble that contrastingly

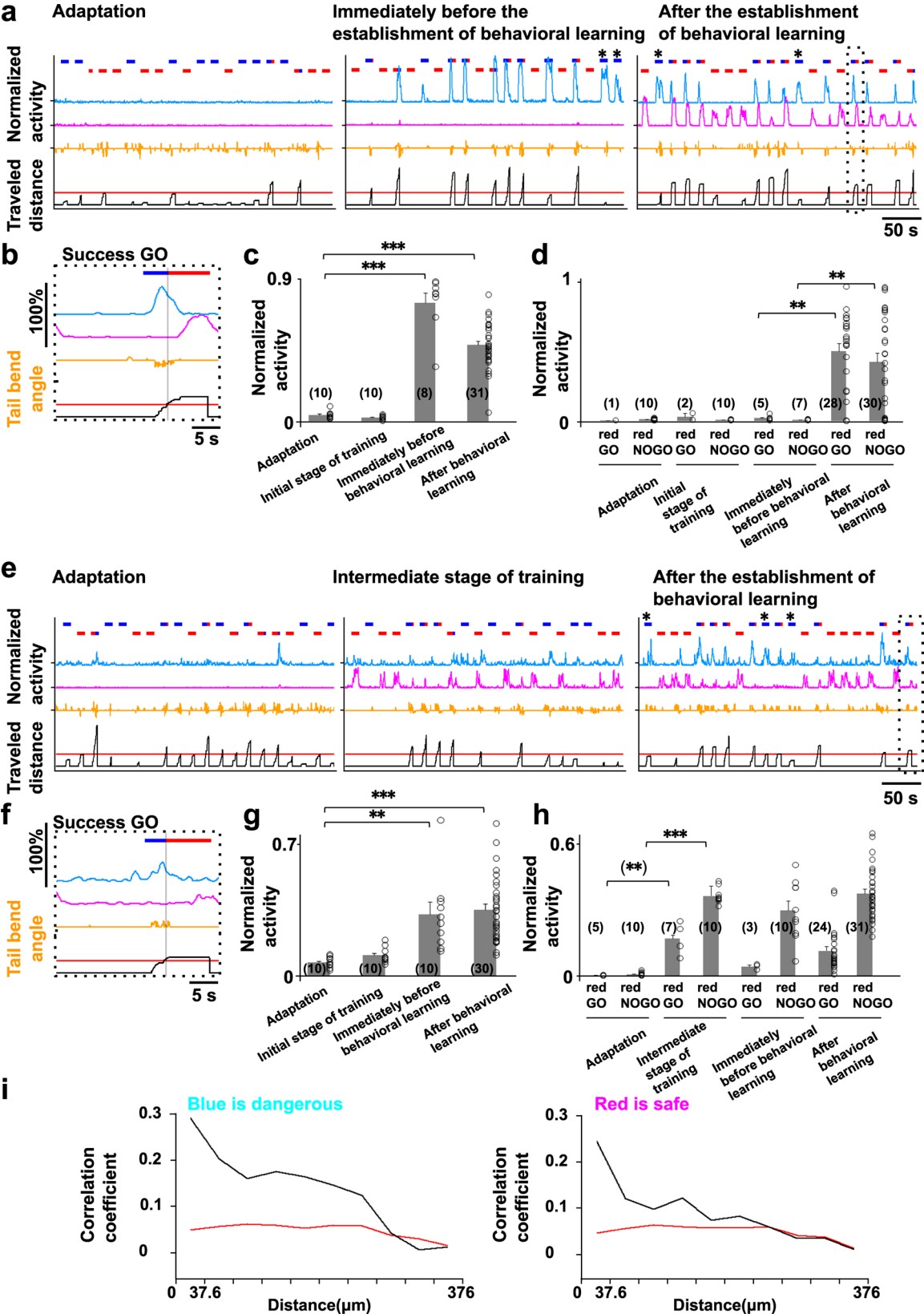

increased in activity when fish reached the red-color goal (Fig. 4b and Supplementary Figs. 5–7a, magenta line, red is safe). This type of ensemble also showed increased activity in the NOGO trials in which fish perceived red color as the start color after the establishment of behavioral learning (Fig. 4a, right panel, magenta line and Supplementary Figs. 5–7a, magenta line, red

is safe). No activity of this ensemble was observed in the adaptation and initial stages of training (Fig. 4a, left panel and Supplementary Figs. 5–7a), suggesting that the ensemble did not encode the simple perception of red color. The activity of this ensemble was observed whenever the fish perceived red color, which was not paired with shock, during repeated trials (Fig. 4d).

**Fig. 4 Repeated trials with the original rule with blue associated with shock and red associated with safety generated the ensembles activated by presentation of blue or red.** Notations in the figures are all the same as in Fig. 3b. **a** The activity of two neural ensembles (cyan and magenta lines) in the adaptation stage (left panel) and the stages immediately before (middle panel) and after (right panel) behavioral learning was established. **b** Enlarged view of the activity of the two ensembles in the boxed area in (**a**). The vertical gray line indicates the time point when the fish reached the goal. **c** Comparison of the cyan ensemble's peak activity in the GO trials in different learning stages. Columns and error bars: mean ± SEM. Each circle indicates the value in each GO trial. The number in parentheses is the number of trials used in the statistics. Adaptation vs immediately before behavioral learning, ***$P = 2.60 \times 10^{-16}$; adaptation vs after behavioral learning, ***$P = 6.43 \times 10^{-13}$, $F(3, 55) = 82.85$. One-way ANOVA, Bonferroni's multiple comparison test. **d** Comparison of the magenta ensemble's peak activity when fish perceived red color in GO and NOGO trials in different learning stages. Columns and error bars: mean ± SEM. Each circle indicates the value in each trial. The numbers in parentheses are the number of trials used in the statistics. Red GO before behavioral learning vs Red GO after behavioral learning, **$P = 7.91 \times 10^{-3}$; red NOGO before behavioral learning vs red NOGO after behavioral learning, **$P = 6.48 \times 10^{-3}$, $F(7, 80) = 9.25$. One-way ANOVA, Bonferroni's multiple comparison test. **e-h** Results of the same analysis as (**a-d**) above for another fish. **g** The numbers in parentheses are the number of trials used in the statistics. Adaptation vs immediately before behavioral learning, **$P = 1.94 \times 10^{-3}$; adaptation vs after behavioral learning, ***$P = 2.44 \times 10^{-5}$, $F(3, 56) = 13.02$ One-way ANOVA, Bonferroni's multiple comparison test. **h** The numbers in parentheses are the number of trials used in the statistics. Red GO adaptation vs red GO intermediate stage, $P = 0.295$; red NOGO adaptation vs red NOGO intermediate stage, ***$P = 9.085 \times 10^{-11}$, $F(7, 88) = 31.67$. One-way ANOVA, Bonferroni's multiple comparison test. Red GO adaptation vs red GO intermediate stage, **$P = 1.42 \times 10^{-3}$. Two-tailed unpaired $t$-test. **i** Relationship between the correlation coefficient and distance among the 10 most-contributing neurons in the color rule-encoding ensembles (left, blue is dangerous rule; right, red is safe rule). The data were averaged from the fish with these ensembles. Black line denotes the averaged correlation from all fish. Red line denotes the average of averaged 10 shuffled data from the fish (see "Methods").

In 17 of the 32 fish (Supplementary Table 1, (+) in both "blue is dangerous" and "red is safe" columns), we observed both ensembles. In 11 of these fish, emergence of the blue responsive ensemble (cyan line) was followed by emergence of the red responsive ensemble (magenta line) as training proceeded (Fig. 4a, b and Supplementary Figs. 6 and 7a). In the remaining six fish, the two ensembles emerged in reverse order (Fig. 4e–h and Supplementary Fig. 5a).

In these blue and red responsive ensembles, approximately 20% of the total imaged neurons showed larger than zero contribution within the ensemble (Supplementary Fig. 8). We observed a similar tendency in the relationships of the activity of individual neurons within an ensemble to the activity of the whole ensemble as in the blue perception-coding ensemble. Namely, even the neurons with lower levels of contribution showed relatively high correlation coefficients to the ensemble activity with intermittent coincidental activation with the ensemble (Supplementary Figs. 5–7b, 2nd and 3rd columns from the left show the results of the ensembles encoding the rule that blue is dangerous and the rule that red is safe, respectively). These neurons tended to be localized at a closer distance with highly correlated activity (Fig. 4i; for individual fish, please see Supplementary Figs. 5–7d), although these ensembles did not have the particular brain region where the neurons within the ensembles of the same nature accumulated across different fish. Supplementary Figs. 10 and 11 show the distribution of the neurons within the two color rule-encoding ensembles, i.e., blue is dangerous and red is safe.

These ensembles showed increased activity depending on the color that was paired with shock or not. The results tempted us to test the possibility that these ensembles encoded rules assigned to the color in these tasks. To address this, we reversed the rules. After behavioral learning was established in a condition wherein blue color was paired with shock and red was safe (original rule), we reversed the rule such that the red color was paired with shock and blue was safe (reversed rule; Fig. 5a). Figure 5b presents the learning curve of GO and NOGO trials under both rules. Immediately after the rule reversal, the fish tended to receive a shock and appeared to freeze (Fig. 5b). Due to this freezing behavior, the apparent success rate of NOGO in the following trials transiently increased and then decreased again, but that of GO remained low. However, as training proceeded, the fish gradually managed to swim forward in the GO trials, and both the GO and NOGO success rates met the learning criteria under the reversed rule. This suggests that learning is indeed required to meet the learning criteria, even in NOGO trials, although this was not evident under the original rule. Eighteen of the 52 tested fish learned the original rule, but only seven of 18 also learned the reversed rule. Among these seven fish, five fish had sufficiently little displacement along the $z$-axis in their imaging data for analysis. In four of these five fish, in the original rule, we could identify the ensemble that increased activity when the fish perceived blue color with repeated trials, but not in the adaptation and initial stages of training (Fig. 5c, cyan line and 5d, left panel). In two of these four fish, the trials in the reversed rule were continued, even after the learning criterion was met. In this case, the ensemble that was activated when the fish perceived blue color in the original rule (Fig. 5c, upper left panel, cyan line and 5d, left panel) ceased to be activated by blue color presentation when the appropriate behavior was learned in the reversed rule (Fig. 5c, bottom right panel, cyan line and 5d, left panel). In the remaining two fish, the trials in the reversed rule were terminated when the success rate of GO trials in the reversed rule reached 80% (Fig. 5e). In these fish, the activity of the ensemble that originally responded to blue presentation did not disappear fully during the NOGO trials, with blue color presented at the beginning of the NOGO trial, even after behavioral learning was established in the reversed rule (Fig. 5f, cyan line). These fish tended to swim forward erroneously when they perceived blue color in the NOGO trials with the reversed rule, although they still managed to stay within the safe blue region (boxed area of Fig. 5f; compare with the null traveled distance of NOGO trials in Fig. 5c, bottom right panel). This observation supports that the ensemble assigned a rule to the blue color, that blue is dangerous in the original rule, and the effectiveness of this rule assignment may not have waned fully in these two fish, even in the reversed rule.

Another group of fish (eight out of 32 fish) experienced only the original rule but performed trials for a longer period than the group of fish that experienced both rules. The averaged ensemble activities in the last ten GO trials ($L$) in these eight fish were not below the averaged ensemble activities immediately after the establishment of behavioral learning ($I$) ($L/I = 3.25 \pm 0.97$), indicating that the ensemble that showed increased activity upon presentation of blue color in repeated trials continued to be responsive until the end of the experiment period. The continued activation contradicts the possibility that habituation might cause the diminished activity of the ensemble encoding the rule that blue is dangerous after rule reversal.

In these four fish that experienced rule reversal, we observed the other ensemble that showed increased activity whenever fish

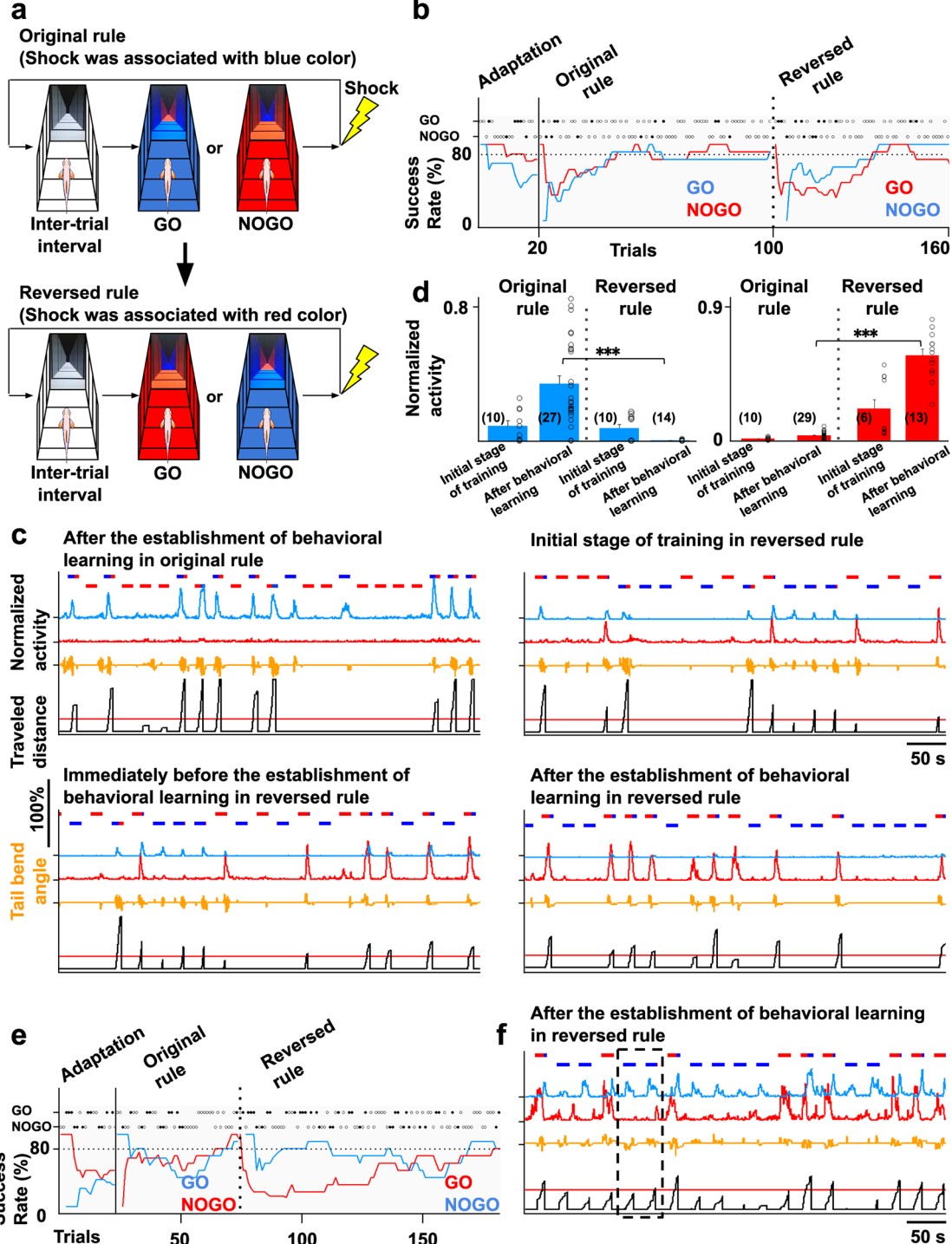

perceived blue color in a manner independent of behavioral learning. This ensemble continued to be activated whenever fish saw blue color, both in the GO trials under the original rule, and in the NOGO trials after rule reversal (Supplementary Fig. 12, blue line), supporting that this ensemble encoding the perception of blue color was not subject to rule reversal.

In the two fish that experienced rule reversal, we also observed the ensemble that got repeatedly activated upon perception of red under the original rule. The activity of this ensemble diminished after rule reversal (Supplementary Fig. 12, magenta line under both original and reversed rules), further supporting our idea that

the ensemble encodes the rule that red is safe. Although we could not conclude from the behavioral data that fish learn the stop behavior in the NOGO task under the original rule, the generation of the ensemble that is likely to encode the rule that red is safe implies that fish might have learned to recognize safety when fish perceived red color.

In three of five fish, we also observed the ensemble that increased in activity when the fish perceived red color, which was paired with shock, as training was repeated in the reversed rule (Fig. 5c, bottom panels, red line). This ensemble previously showed no activity in the trials under the original rule (Fig. 5c,

**Fig. 5 Rule reversal revealed that each ensemble, which is activated when fish perceived each color as training proceeded, encodes the rule assigned to that color. a** GO/NOGO tasks with the original and reversed rules. **b** The learning curve of GO/NOGO trials with the original and reversed rules. Vertical line, initiation of the original rule; vertical dotted line, initiation of the reversed rule. In the adaptation and with the original rule, the success rates of GO and NOGO trials are indicated by blue and red lines, respectively. In the reversed rule, the success rates of GO and NOGO trials are indicated by red and blue lines, respectively. Open circle indicates successful trial. Solid circle indicates failed trial. **c** The activity of two ensembles after behavioral learning was established with the original rule (upper left panel), immediately after rule change (upper right panel), and immediately before (bottom left panel) and after (bottom right panel) behavioral learning was established with the reversed rule. **d** Left: comparison of the cyan ensemble's peak activity in (**c**) when fish perceived blue color as a starting color with the original and reversed rules in the initial stage of training and after behavioral learning. Columns and error bars: mean ± SEM. Each circle indicates the value in each trial. After behavioral learning of original rule vs after behavioral learning of reversed rule, [***]$P = 1.80 \times 10^{-6}$, $F(3, 75) = 14.28$. One-way ANOVA, Bonferroni's multiple comparison test. Right: comparison of the red ensemble's peak activity in (**c**) when fish perceived red color as a starting color with the original and reversed rules in the initial stage of training and after behavioral learning. Columns and error bars: mean ± SEM. Each circle indicates the value in each trial. The numbers in parentheses are the number of trials used in the statistics. After behavioral learning of original rule vs after behavioral learning of reversed rule, [***]$P = 9.73 \times 10^{-10}$, $F(3, 36) = 36.34$. One-way ANOVA, Bonferroni's multiple comparison test. **e** The learning curve of GO/NOGO trials of another fish in which the cyan ensemble did not disappear. Notations are identical to (**b**). **f** The activity of the two ensembles after behavioral learning was established with the reversed rule. The cyan and red lines indicate the ensembles encoding the color rule that blue is dangerous and red is dangerous, respectively.

upper left panel, red line and 5d, right panel), likely encoding an association of the red-color scenery with danger. Altogether, our results supported the hypothesis that these ensembles functioned to assign rules to the colors, i.e., a rule that blue is dangerous, and another rule that red is safe in the original rule.

**A neural ensemble activated in open-loop GO and closed-loop failed GO trials.** Taking advantage of the virtual reality system, we performed an open-loop experiment in all 32 learner fish (Fig. 6a) after behavioral learning was established under the original rule for GO and NOGO tasks in the closed-loop condition. In the open-loop condition, tail movement of the fish was not programmed to induce a backward movement of the scenery, and hence no sensory flow was detected by the fish, and the goal was not reached. Accordingly, in the GO trial in the open-loop condition, the activity of the ensemble that we interpreted to assign a rule that blue is dangerous remained elevated until the end of the GO trials. The fish continued beating the tail as long as the blue color was presented (Fig. 6b, c, cyan line). In contrast, the ensemble that we considered to assign a "red-safe" rule did not get activated in the GO trials (Fig. 6b, c, magenta line), confirming that the perception of the color-induced activation of the color-associated rule-coding ensembles.

Under the open-loop condition, as shown in the green line in Fig. 6d, we noticed another type of ensemble that exhibited continuously enhanced activity throughout the GO trials in the open-loop condition in 10 of the 32 fish (Fig. 6d, upper left panel, 6f; all data of this fish are summarized in Supplementary Table 1 and Supplementary Fig. 7). Unlike the previously described ensemble encoding the perception of blue color, this ensemble was not activated in the successful GO trials after behavioral learning was established in the original closed-loop condition (Fig. 6d, upper right panel, 6e, right panel, 6f, and Supplementary Figs. 5–7a, scenery flow prediction error (SFPE)). In nine of these 10 fish, this ensemble got activated specifically in the failed GO trials in the closed-loop condition (Fig. 6d, upper right panel, 6e, left panel, and Supplementary Figs. 5–7a, SFPE). In the GO trials in the open-loop condition, the fish beat their tails vigorously, while in the failed GO trials, the fish did not beat their tails. However, both in the GO trials in the open-loop condition and the failed GO trials in the closed-loop condition, the fish was not presented with the backward moving scenery upon the presentation of blue color.

In summary, we observed that this neural ensemble was activated during the GO task in the open-loop condition and in the failed GO trials, but not in the successful GO trials, under the original rule. In both situations in which this ensemble became

activated, the scenery did not move, although it should have moved backward if the fish had successfully swum forward to avoid receiving a shock.

As this ensemble was also activated in the failed GO trials in which the fish did not beat their tails (Fig. 6e, left panel, orange line), this ensemble was not simply copying motor commands to beat the tail. In the inter-trial interval of this virtual reality with white-color scenery, we switched off the feedback of the tail beats to the virtual reality. This ensemble did not become activated even when the fish beat their tails under this situation, which further supports the notion that the ensemble was not simply copying the motor commands.

In summary, this particular ensemble was activated when the scenery did not flow backward upon blue color presentation when the backward flow of the scenery should be perceived by the fish as long as the fish responded appropriately by swimming forward to avoid receiving shock. This suggest that this ensemble is likely to encode the discrepancy between the real scenery perceived by fish and the scenery predicted by fish as favorable. Hence, we call this the ensemble putatively encoding the SFPE.

In five of the nine fish, this ensemble began to emerge after behavioral learning was established (Fig. 6d–f and Supplementary Figs. 6 and 7a, SFPE), but, in the remaining four fish, the activation of this ensemble was observed preceding the establishment of behavioral learning (Fig. 6g–i and Supplementary Fig. 5a, green line, SFPE).

In this ensemble, approximately 20% of the total imaged neurons showed larger than zero contribution within the ensemble (Supplementary Fig. 8). These ensembles also showed a similar tendency in the relationships of the activities of individual neurons within an ensemble to the activity of the ensemble itself as we observed in the blue perception-coding ensemble. Namely, even the neurons with lower levels of contribution showed relatively high correlation coefficients to the ensemble activity with intermittent coincidental activation with the ensemble (Supplementary Figs. 5–7b, right column and Supplementary Figs. 5–7c, bottom right panel, SFPE). These neurons tended to be localized at a closer distance with highly correlated activity (Fig. 6j; for individual fish, please see Supplementary Figs. 5–7d, bottom right panel, SFPE) although these ensembles did not have the particular brain region where the neurons within the putative SFPE-encoding ensemble accumulated across different fish. Supplementary Fig. 13 shows the distribution of the neurons within the putative SFPE-encoding ensemble across different fish.

We tested an alternative interpretation of the role of the ensemble originally assigning a rule that blue is dangerous in GO trials. The fish may expect that perceiving the red environment is

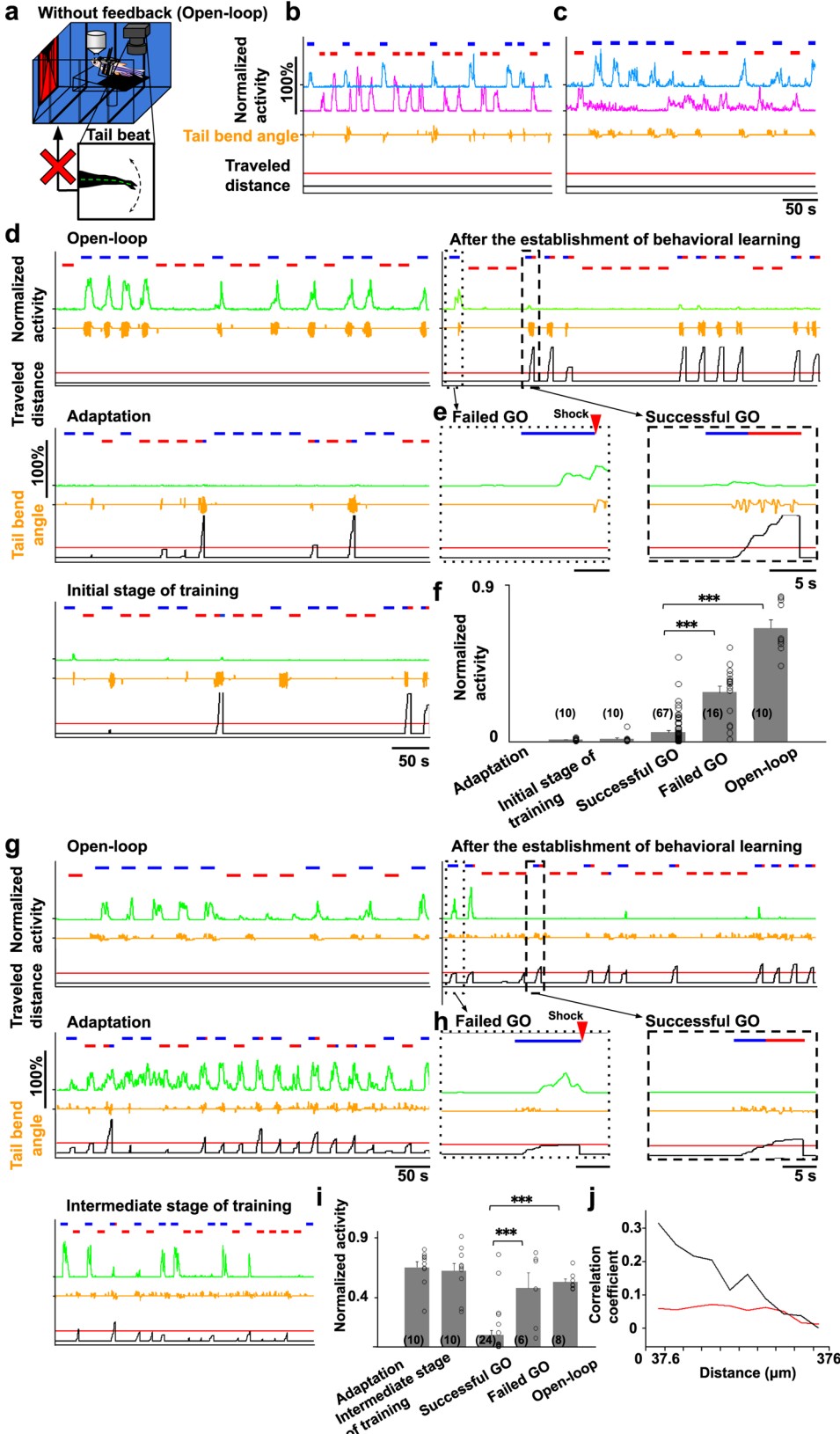

favorable for future safety, and this ensemble may also encode a prediction error between such expectation of future preferable color (red) and real perceived color (blue). However, our experimental results did not support this hypothesis as explained in detail in the legend of Supplementary Fig. 14, which shows the results of the goal color change experiment.

**Fish with a putative SFPE swam straight toward the goal**. As mentioned above, we observed the ensemble assigning a rule that blue is dangerous in 24 of the 32 fish (Fig. 4 and Supplementary Table 1). Only eight of these 24 fish also had another ensemble putatively encoding the SFPE (Fig. 6 and Supplementary Table 1,

**Fig. 6 A specific neural ensemble got activated by the absence of perception of visual backward movement upon blue color presentation.** Notations in the figures are identical to Fig. 3b. **a** The open-loop experiment in which feedback was turned off. The scenery did not move in response to the tail beat. **b**, **c** The activity of the two ensembles encoding the color rules that blue is dangerous (cyan line) and that red is safe (magenta line) in the open-loop condition. **b** Data from the fish used in Fig. 4a–d. **c** Data from the fish used in Fig. 4e–h. **d** The activity of an ensemble in the open-loop condition (upper left panel), after behavioral learning was established (upper right panel), in the adaptation (middle left panel), and in the initial stage of training (bottom left panel). The activity of the ensemble increased in the GO trial in the open-loop condition and the failed GO trial but not in the successful GO trials in the closed-loop condition. **e** Left: enlarged view of a failed GO trial (boxed area of the dotted line in (**d**) upper right panel). Right: enlarged view of a successful GO trial (boxed area of the dashed line in (**d**) upper right panel). Red rectangle indicates the timing of shock. **f** Comparison of peak activity of the ensemble when fish perceived the starting color in the adaptation stage, initial stage of training, successful GO trials, failed GO trials, and open-loop GO trials. Columns and error bars: mean ± SEM. Each circle indicates the value in each GO trial. The numbers in parentheses are the number of trials used in the statistics. Successful GO vs failed GO, ***$P = 7.79 \times 10^{-13}$; successful GO vs open-loop, ***$P = 2.54 \times 10^{-34}$, $F_{(4, 110)} = 101.97$. One-way ANOVA, Bonferroni's multiple comparison test. **g–i** Results of the same analysis as (**d–f**) above for another fish. The timing of the increased activity along with behavioral learning was different from the data shown in (**d–f**). **i** The numbers in parentheses are the number of trials used in the statistics. Successful GO vs failed GO, ***$P = 5.02 \times 10^{-4}$; successful GO vs open-loop, ***$P = 8.93 \times 10^{-6}$, $F_{(4, 53)} = 23.86$. One-way ANOVA, Bonferroni's multiple comparison test. **j** Relationship between the correlation coefficient and distance among the 10 most-contributing neurons in the putatively SFPE-encoding ensemble. The data were averaged from 9 fish with this ensemble. Black line denotes the averaged correlation from 9 fish. Red line denotes the average of averaged 10 shuffled data from 9 fish (see "Methods").

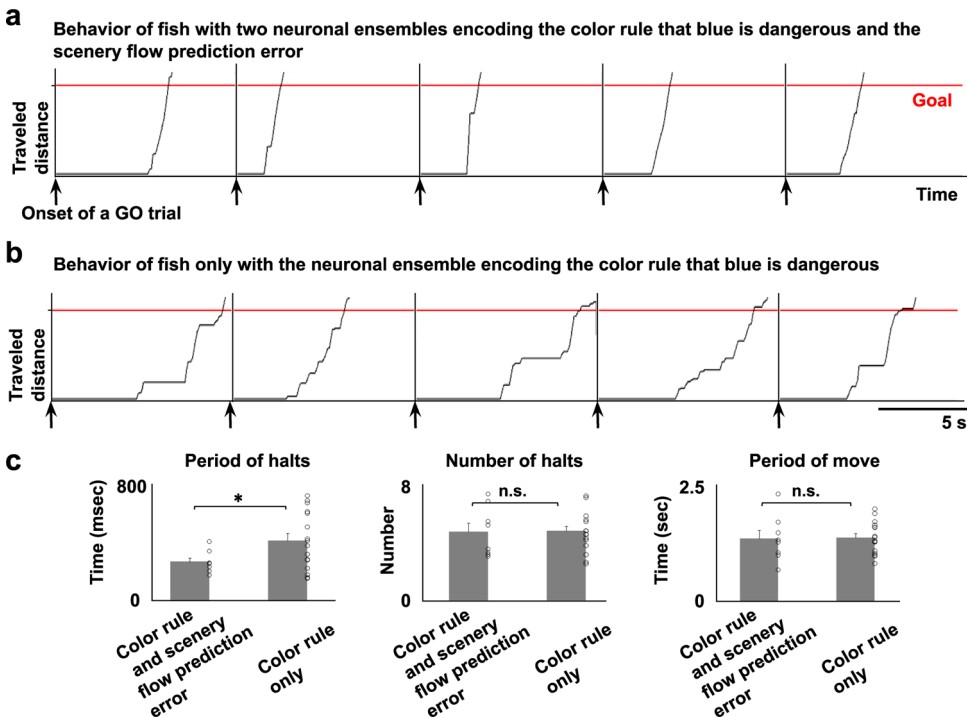

**Fig. 7 Fish with the scenery flow prediction error ensemble swam straight toward the goal with shorter halts than fish with only the color rule-coding ensemble. a**, **b** The traveled distance in five successful GO trials after the establishment of behavioral learning in a fish with both ensembles encoding the color rule that blue is dangerous and the scenery flow prediction error (**a**) and a fish with only the ensemble encoding the color rule that blue is dangerous (**b**). The black line indicates the traveled distance. The red line indicates the goal. **c** Comparison of the period of halts (left panel), number of halts (middle panel), and period of movement (right panel) between two groups. The number of fish which has color rule-coding and putative SFPE ensembles is 8 and that of fish which has only color rule-coding ensemble is 16. Columns and error bars: mean ± SEM. Each circle indicates the value in each fish. Left: *$P = 0.028$ (Cohen's $d = 0.978302$). Middle: $P = 0.455$. Right: $P = 0.457$. One-tailed unpaired *t*-test, n.s., not significant.

(+) in both "blue is dangerous" and "scenery flow prediction error" columns). These data suggest that fish can succeed in the GO trials, even with only the ensemble assigning a rule that blue is dangerous. This result made us question the role of the additional ensemble putatively encoding the SFPE in behavioral control.

To address this, we first compared the swimming patterns in the successful GO trials of the fish with both ensembles and the fish with only the ensemble assigning a "blue-dangerous" rule to identify any difference in their swimming patterns (Fig. 7a, b and Supplementary Data 1).

Intriguingly, the sum of the halt time en route to the goal was significantly shorter in the fish with both ensembles (Fig. 7c, left

panel). The two groups did not differ in the number of halts and the period of movement until the fish reached the goal (Fig. 7c, middle and right panels).

The putatively SFPE-encoding ensemble was observed in eight out of 24 fish that possessed a color rule-encoding ensemble. We examined the possibility that the failure to detect the putatively SFPE-encoding ensemble in the remaining 16 fish might result from the limitation in the scanning volume of the telencephalon. To address this question, we extended imaging into a deeper telencephalic region by increasing the number of imaging planes from three to six (Supplementary Fig. 15a shows the averaged calcium images of six planes). NMF calculations were performed

in the deeper three planes and the superficial three planes separately and compared. In one out of four learner fish, we identified the putatively SFPE-encoding ensemble. Consistent with the previous results, this fish showed shorter halt periods than the fish with only the color rule-encoding ensemble (Supplementary Fig. 15b). Supplementary Fig. 15b shows the halt period of the two groups. The putatively SFPE-encoding ensemble was observed in the surface planes but not in the deeper planes (Supplementary Table 2 summarizes results of identified ensembles in surface and deeper planes in the four fish), implying that the surface three-plane imaging might be enough to capture the putatively SFPE-encoding ensemble.

The permutation test of randomly dividing all 28 fish into two groups of nine and 19 fish revealed that the difference of halt periods between two groups randomly made for 1000 times rarely exceeded the actually observed difference between the two groups categorized by the presence or absence of a putatively SFPE-encoding ensemble (permutation $P$-value = 0.011 < 0.05), confirming that the shorter halt period of fish with both color rule and putatively SFPE-encoding ensembles was not accidental.

Therefore, this statistical analysis confirmed that the fish with both ensembles swam forward more efficiently, i.e., swam straight toward the goal, than the fish with only the ensemble assigning a "blue-dangerous" rule, which paused for a longer period on the way to the goal (Fig. 7a, b and Supplementary Data 1).

**Computational models replicate behavioral differences.** Based on the aforementioned empirical observations, we developed a neural network model of fish. We supposed that the input from the blue perception neurons (BP neurons; Fig. 8a, blue circle) activated downstream neurons that generate motor signals to make fish swim forward (SF neurons; Fig. 8a, red circle). The strength of the input was updated through an activity-dependent plasticity regulated by signals of two ensembles encoding the reward prediction error (RPE) and SFPE (Fig. 8a, green and cyan circles). We further supposed that the SFPE ensemble computed the difference between the bottom-up backward flow signals and the top-down backward scenery flow prediction (SFP); whereas the SFP ensemble was self-organized through plasticity mediated by the RPE ensemble. This model could simulate the emergence of the SFP and SFPE ensembles in the network through an association between visual perception and punishment—consistent with our empirical observations of the emergence of the SFPE ensemble in the training stage (Fig. 8b; compare with Fig. 6d).

Using this computational model, we examined whether our observation—i.e., the straight swimming pattern of the fish with the SFPE ensemble—could be explained by the hypothesis that the SFPE played a role in facilitating taking optimal action to minimize the prediction error between the real observed scenery and the predicted backward moving scenery.

At each time point on its way to the goal during blue color presentation, we can assume that the fish makes a decision whether to go forward or stop depending on how strongly the input from the BP neurons can activate downstream neurons that make fish swim forward (i.e., SF neurons). If the aforementioned hypothesis is true, every time the fish stops en route to the goal during blue color presentation and the backward flow of the scenery is interrupted, the fish would experience the activation of the SFPE ensemble. The SFP ensemble would expect its presence, inducing the SFPE signal. Here, the SFPE signal would act to strengthen the connection between the BP and SF neurons. This would hinder the fish from stopping and favor a GO choice with a gradually higher probability, because the SFPE can be minimized by this behavioral choice. After repetition of this process, the fish would eventually learn to swim straight to the safe goal without choosing to stop at any point on its way to the goal. This swimming pattern is consistent with our observations, and the simulated fish model could replicate empirically observed fish behaviors (Figs. 7a and 8a, c). These results support the role of the SFPE ensemble in the strengthening of the connectivity between BP and SF.

In contrast, the fish with only the ensemble assigning rules to colors learn the escape behavior according to the naive reinforcement learning, as we described previously[9]. Here, receiving punishment by staying in the blue region brings about an expected reward level, and avoiding punishment by moving into the red region generates a positive RPE. In this case, the connectivity between the BP and SF neurons is strengthened relatively infrequently, i.e., only when the fish happens to successfully reach the safe red goal. Moreover, the strengthening of this connectivity ceases at a relatively early stage of the training; as soon as it reaches a level enough to bring the fish to the goal within the given time limit before shock is given. With this still relatively low level of connectivity, the activation of the BP neurons cannot always activate the SF neurons. Thus, the fish would keep the tendency to stop on its way to the goal, even after they become able to constantly reach the safe goal (Figs. 7b and 8a, d). Hence, we could attribute the mechanisms underlying two different swimming strategies we observed to the facilitation of the connectivity between the BP and SF neurons induced by the SFPE ensemble.

## Discussion

In this study, we trained adult zebrafish in closed-loop and open-loop virtual reality environments for active and passive avoidance, and analyzed the activities of neural populations in the wide surface area of the telencephalon by using the unsupervised method, NMF. We could successfully demonstrate that such training induced the generation of two types of neural ensembles, one encoding rules attached to a specific color and the other putatively encoding the SFPE (Fig. 9). We addressed the problem of multiple comparison to exclude the possibility that these ensembles were identified by chance. For this purpose, we created the semi-automatized selection method and applied it to the NMF ensembles generated from the neural population with randomly shuffled neural activity (for details please see "Methods" and Supplementary Figs. 16 and 17). As a result, no ensemble was found to meet the criteria in the automatized selection method (Supplementary Fig. 17), suggesting that the ensembles we selected were not obtained by chance.

The capacity of zebrafish to generate putative SFPE demonstrates that zebrafish are capable of predictive coding of the favorable state based on the internal model of the outer cognitive world, as exemplified in other higher vertebrates[25,29–36], although it may be rudimentary, to derive the state prediction error by comparing reality with prediction[2,3].

Importantly, fish with only the ensemble encoding a rule assigned to the surrounding color managed to reach the safe region by intermittent swimming on their way to the goal. In contrast, fish with both the color rule-encoding and the putative state prediction error-encoding ensembles showed a more efficient swimming pattern, i.e., straight swimming toward the goal without rest.

This behavioral difference, taken together with computational modeling, supports that the fish can utilize the putative SFPE ensemble for efficient behavioral optimization by trying to minimize the state prediction error (surprise), i.e., the discrepancy between real sensory inputs and model-based sensory predictions (Fig. 9). Nonetheless, further experiments are necessary to unequivocally prove this hypothesis, such as using targeted optogenetic manipulation to change the activities of the neurons

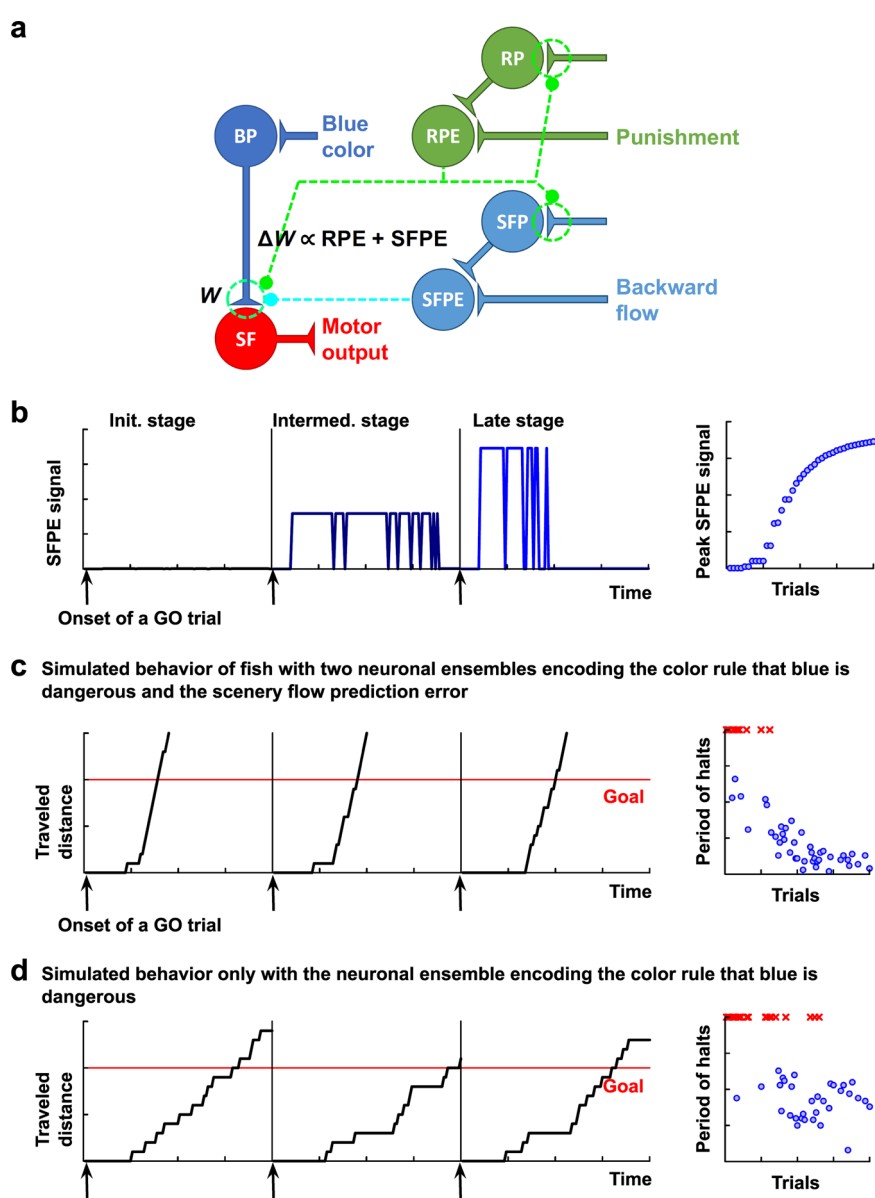

**Fig. 8 Computational models replicate behavioral differences. a** Schematic circuit diagram to select swimming forward. The neural network comprises ensembles encoding the perception of blue (BP, blue) and swimming forward (SF, red). Moreover, two additional units represent mutually different prediction errors. The reward prediction error (RPE, green) ensemble computes the difference between the (negative) reward prediction (RP) and actual punishment. Whereas, the scenery flow prediction error (SFPE, cyan) computes the difference between the scenery flow prediction (SFP) and actual backward flow. Here, the RPE ensemble took a positive value when the fish could avoid the punishment contrary to its expectation; otherwise, it took zero. The SFPE ensemble self-organized in the early stage of training to take a positive value when the fish sensed the SFPE at a given time point or take zero otherwise. The synaptic potentiation of $W$ occurred only when the fish detected those errors, leading to minimization of those errors. **b** Emergence of the SFPE activity. Left: time-lapse activity changes in the three GO trials in the initial (left), intermediate (middle), and later (right) stages of training. Right: learning curve depicting the increase in activity of the SFPE ensemble. First 40 GO trials are shown. In the model, this increase occurred as a consequence of the emergence of an ensemble that encodes scenery flow prediction, because the SFPE ensemble is activated when the prediction exceeds the actual scenery flow sensation. **c** Behavior of synthetic fish in the presence of the RPE and SFPE-coding ensembles. Left: trajectories in the last three GO trials. The red line indicates the goal position. Right: the period of halts in GO trials over the training period. Because the fish sensed the SFPE whenever it stopped in the blue colored area, the SFPE signal potentiated the synapse more efficiently than the RPE alone, leading to behavior involving reaching the goal with shorter halts. **d** Behavior of synthetic fish in the RPE-coding ensemble. Left: trajectories in the last three GO trials. The red line indicates the goal position. Right: the period of halts in GO trials over the training period. The blue circles and red crosses denote successful and failed GO trials, respectively.

in this ensemble. The process of improving the goal-directed behavior by minimizing the state prediction error, observed in fish with two ensembles, was consistent with a theoretically advocated fundamental mechanism underlying human and animal behavioral control termed active inference[2,3]. This contrasted significantly with the outcome of naive reward value maximization observed in fish with only one ensemble.

Whether animals adopt the reward value maximization and/or the surprise minimization principle for goal-directed behavior in real life has been the subject of debate[4,5]. We showed here that

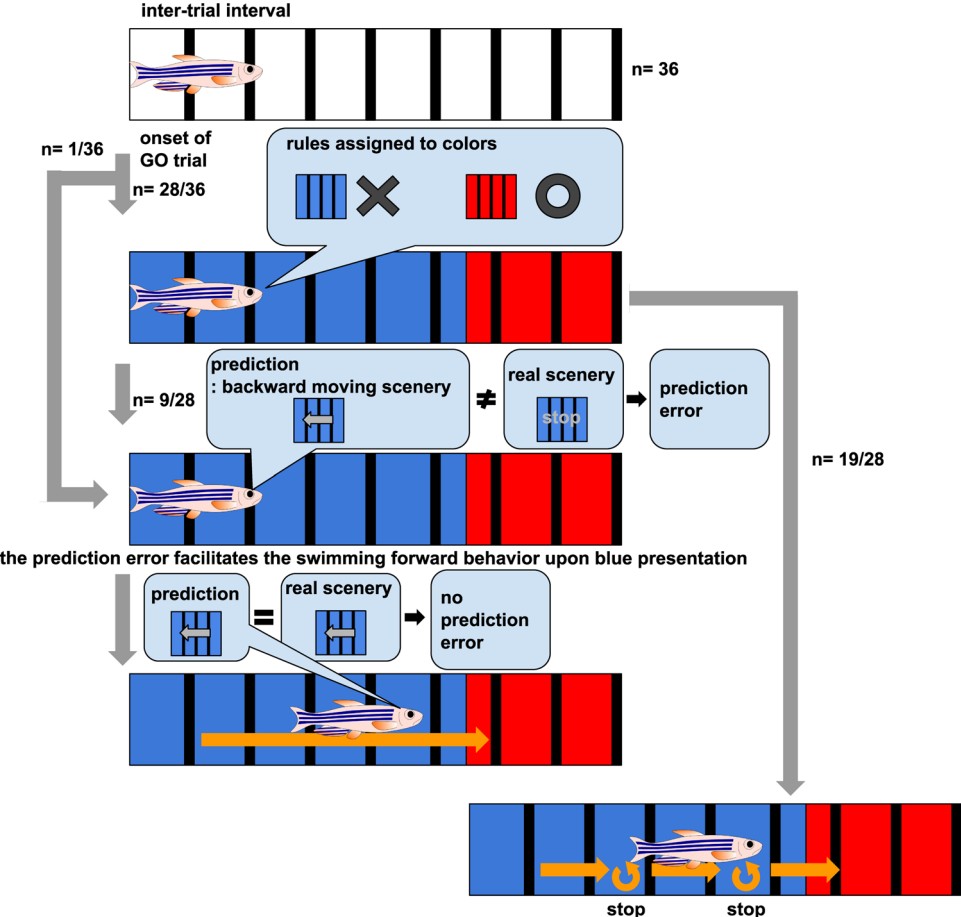

**Fig. 9 Schematic illustration showing the relationship between the generated neural ensembles and behaviors.** Adult zebrafish formed a cognitive map by assigning rules to presented colors and generated the ensemble putatively encoding the scenery flow prediction error (SFPE). Activation of the putative SFPE ensemble improved active avoidance behavior.

zebrafish in fact can adopt either the former alone or both principles, leading to different consequences in behavioral control.

With regard to the two ensembles associating a "blue-dangerous" rule (anticipation of danger) to blue color and a "red-safe" rule (anticipation of safety) to red color (Fig. 9), the generation of these ensembles preceded the establishment of correct behaviors. The activation of this ensemble, when sensing blue color, was observed even in the failed GO trials with the original rule. These observations suggested that the process of learning to assign rules (danger or safety) to the cognitive inputs (blue or red) is separate from learning appropriate behaviors to achieve the maximum reward (safety). This two-step learning process differs from conventional model-free reinforcement learning, which directly attaches values to motor outputs for the selection of best-rewarding behavior[1,37,38]. In mammals, the prefrontal cortex has been implicated in value assignments[39]. Whether it imparts values to cognitive inputs, or to motor outputs or to both, has been debated[40,41]. Our results show that these processes are separable in goal-directed behaviors.

Recently, Huang et al. developed another closed-loop virtual reality system for freely swimming fish independently from us, and observed in the telencephalon a population of neurons that were activated only when there was a mismatch in the visual feedback in response to fish movement[36]. Similar neural activity was also observed in the visual cortex of mice[31,32]. These activities would encode the error between the visual input and the predicted image converted from the efference copy of the motor

output by the forward model[4,5]. In contrast, the ensemble putatively encoding the SFPE got activated independently of the motor efference copy, because the putative SFPE was activated in failed GO trials wherein fish did not beat their tails.

We identified the neural ensemble putatively encoding the SFPE only in one-third of the fish that successfully learned active avoidance. It is unclear why only a part of the fish could generate this ensemble. Fish with, and fish without, this ensemble required a comparable number of trials until behavioral learning was established (Supplementary Fig. 21), suggesting it is unlikely that only the fish requiring more trials to learn escape behavior eventually generated this additional ensemble.

Putative SFPE signals are expected to be generated by the mismatch between top-down signals that encode predicted sensory states and bottom-up signals conveying the actual sensory states and are canceled out by bottom-up signals when the real states fit with the predicted states[42]. Therefore, to reveal how neural ensembles self-organize to encode the putative SFPE, future work will need to discriminate the activities of both excitatory and inhibitory neurons and analyze how they interact with each other in computing state prediction errors.

In summary, our results propound that zebrafish are capable of assigning rules to the scenery they see, and of generating a state prediction error by comparing reality with a prediction derived from an internal model[2,3]. Observed refinement of the escape behavior by fish with state prediction error provided supporting evidence for the hypothesis that this future state prediction error

is indeed used for behavioral optimization as proposed in the active inference theory[2,3].

The ensemble activated by the perception of blue exhibited significant overlap in neural populations with the ensemble assigning a rule that blue is dangerous and with the ensemble putatively encoding the SFPE (Supplementary Fig. 18). This suggests that these overlapping neurons were not simply involved in color perception but, after several trials, became activated again in the process of cognitive rule assignment to the perceived scenery (colors) and to the derivation of future prediction error.

We found that paired neurons with a high correlation coefficient in their activity are located at a closer distance when we focused on the 10 most-contributing neurons in each ensemble (Figs. 3g, 4i, and 6j), implying the generation of a localized unit of neurons that encode similar information in the telencephalon of adult zebrafish. Similarly, closely located neurons showed similar activity patterns in the striatum, the downstream target of cortical excitatory neurons in mice[43]. In zebrafish, this cortico-striatal projection exists and is considered to play an important role in decision-making[10,13]. Further study is necessary to reveal the relationship between these localized units in the dorsal pallium and their downstream target to elucidate the mechanisms of behavioral choice.

In this study, we reported ensembles encoding blue perception, the rule that blue is dangerous, the rule that red is safe and putative SFPE. Although a majority of the fish generated each of these ensembles except the one putatively encoding the SFPE (Supplementary Table 1), some fish lacked some of these ensembles. This raises the possibility that we might have missed these ensembles because our imaging did not cover the entire telencephalon. Indeed, we still missed the ensemble encoding perception of blue in one out of four fish, even in the larger volume imaging (Fig. 1c, red rectangle region and Supplementary Table 2). In the case of the putatively SFPE-encoding ensemble, we found this ensemble only in the surface region of the telencephalon, even when imaging a larger volume (Fig. 1c, red rectangle region and Supplementary Table 2), implying the possibility that the putative SFPE was represented in the surface region of the telencephalon, although further study for imaging the whole telencephalon is necessary to confirm this point.

Although we observed neural activity in a relatively wide region of the dorsal telencephalon, we could not cover the entire dorsal telencephalon. In fact, although electrophysiology studies in percomorphs revealed the visual sensory input to the dorsal part of the Dl region from the preglomerular nucleus, suggesting the equivalence of this region to the primary visual cortex in mammals[44], we could not fully image this area in this study. This may be why we missed the ensemble encoding blue perception in some fish. Further experiments for capturing neural activity in the wider and thicker areas of the telencephalon are necessary to address this limitation.

As with many cases of adaptive behavioral learning in mammals, the fish evolutionary equivalents of the mammalian striatum and globus pallidus in the basal ganglia, the dorsal subpallium and entopeduncular nucleus, could also be involved in assigning cognitive rules and in learning correct behaviors based on future prediction. In addition, the zebrafish brain has neuro-modulatory systems, such as those of dopamine and serotonin as in the mammalian brain[9,45,46]. All these structures play important roles in decision-making behaviors.

The small size of the zebrafish telencephalon will allow us in the future to observe the coordinated in vivo neural activity of these brain areas simultaneously by applying advanced technologies for wide, deep, and fast imaging, such as advanced laser scanning methods[47,48], three-photon microscopy[47,49], and adaptive optics[50,51].

## Methods

**Animals**. All surgical and experimental procedures were reviewed and approved by the Animal Care and Use Committees of the RIKEN Center for Brain Science. Zebrafish (*Danio rerio*) were bred and raised under standard conditions[52]. In this study, we used *TgBAC(camk2a:GAL4VP16)^rw0154a*; *TgBAC(vglut2a:Gal4)*[53]; *Tg(UAS:G-CaMP7)* *^rw0155* zebrafish aged >6 months in the nacre[18] or casper[19] background. We used camk2a containing BAC clones zH278O8 to establish *TgBAC(camk2a:GAL4VP16)* as reported previously[9] and the *UAS:G-CaMP7*[54] plasmid provided by Drs. Koide and Yoshihara[55].

**Virtual reality**. The virtual reality environment consisted of four LCDs (Good display, Model No. GD70MLXD), which presented the visual stimuli. In this environment, fish can swim and stop along a one-dimensional track consisting of a white, blue, or red background color with black stripes perpendicular to the direction of swimming. The tail movement was captured by a web camera (BSW20KM11BK, iBUFFALO) with some modifications i.e., the filter was replaced with the infrared sharp cut filter (IR 76 FUJI FILTER, FUJIFILM) to avoid inter-ference from the display light. The tail was illuminated with infrared LED light (850 nm; ILR-IO16-85NL-SC201-WIR200, intelligent LED solutions). The virtual reality system was controlled by custom-written programs based on LabVIEW (National Instruments), Matlab (MathWorks), and OMEGA SPACE (Solidray Co. Ltd.). The virtual reality environment was created using OMEGA SPACE software, which provides built-in tools for the creation of virtual space. As the fish was fixated inside the tank, it could move its tail, and the tail beat frequency was continuously detected to cause forward-only motion in the virtual space. We used the following method to investigate the frequency. The body of the fish was detected using a camera placed at the top of the tank, and a custom program based on Matlab Computer Vision Toolbox and LabVIEW. The camera was placed in such a manner that the body of the fish was along the vertical (Y) axis of the image taken by the camera. For each time frame of 100 ms, a centerline lying along the body was calculated by connecting the midpoints of the body in every row of the image. The upper points of the centerline corresponded to the upper portion of the body. The uppermost portion of the body in view did not move during tail-bending; thus, we took the five uppermost points of the centerline and calculated a straight line, which was at 90° of inclination to the horizontal axis of the image. We then took the five lowermost points in the centerline and fitted a straight line comprising these. When the fish bent its tail, this second line created an angle of inclination to the initial straight line. When the fish did not move its tail, this angle was 180°, and the angle was reduced when the fish moved its tail. Thus, we subtracted this angle from 180° to determine how much the tail shifted from the reference position. This subtracted value was used as the tail beat angle. Furthermore, we observed that there was bias in the direction of the tail angle depending on the fish; we therefore took a com-paratively large moving time window of 10 s and averaged the tail beat angles in that window to set the reference angle of the tail. In each time step, the difference between the reference and the tail beat angle was calculated to obtain the actual tail bend angle, which was used for the calculation of tail beat frequency. To calculate the frequency, we performed a Fourier transform using the current and nine pre-vious values of the tail beat angle, corresponding to 1 s of length in our system. The calculated frequency was multiplied by the arbitrary gain constant to use as the forward speed of the fish in the virtual reality space, as the speed of the fish was considered to have a linear correlation with the tail beat frequency[56].

**Fixation of living adult zebrafish**. To capture the tail movement and image the telencephalic neural activity, fish were tethered to the custom-made harness. Adult zebrafish were briefly anesthetized with 0.02% tricine (ethyl 3-aminobenzoate methane sulfonate salt, Sigma-Aldrich) diluted in fish rearing water and mounted on a hand-made surgery apparatus. During surgery, fish rearing water with 0.02% tricaine was continuously perfused to keep fish alive and maintain anesthesia. First, the skin above the skull over the telencephalon and the tectum was removed using micro knives (10315-12, Fine Science Tools; Supplementary Fig. 1a1). After drying the surface, the dental bond (Scotchbond Universal Adhesive, 41255, 3I) was pasted on the skull using a toothpick and illuminated with blue LED light for 10 s (Supplementary Fig. 1a2). A U-shaped metal was placed on the skull (Supple-mentary Fig. 1a3) and then fluidic dental cement (Filtek Supreme Ultra, 6032XW, 3I) was placed on the skull (Supplementary Fig. 1a4). Blue LED light was used to harden the cement and fix the U-shaped metal on the skull (Supplementary Fig. 1a4). The base of the fixation apparatus, the assembled harness, and the plastic ceiling to hold the fish body (Supplementary Fig. 1a5) were fixed (Supplementary Fig. 1a6). The tips of the U-shaped metal were inserted into the slits in the small part of assembled harness (Supplementary Fig. 1a7), additional dental cement was placed on the contact point between them, and blue LED light was applied for 10 s (Supplementary Fig. 1a8). After recovery from the anesthesia by perfusion with fish rearing water, the tethered fish and the fixation apparatus were transferred to the tank with the virtual reality environment (Supplementary Fig. 1a9). The tethered fish was kept in the dark condition for ≥30 min under the microscope to allow it to habituate to the tethered situation with continuous perfusion of fish rearing water.

**GO/NOGO tasks**. Fish were trained for the GO task (active avoidance) and the NOGO task (passive avoidance) in the virtual reality environment under the

microscope. Prior to the training, fish were kept separated in individual 1 L tanks overnight. After fixation to the custom-made harness followed by habituation to the tethered state, visual stimuli from four displays started to be presented and the adaptation session for visual stimuli was started. This adaptation session consisted of 20 GO/NOGO trials (ten GO trials and ten NOGO trials in random order) without the application of electric shocks. After the adaptation session, we started the training session with electric shocks. One training session consisted of 60 GO/NOGO trials. We mixed ten GO and ten NOGO trials in a random sequence in the first, middle, and last 20 trials. Each fish performed 2–5 sessions. The inter-session interval continued for 15–20 min without the presentation of visual stimuli. In the inter-trial interval in which the feedback of the tail beat to the virtual reality was off, the fish was presented with a white background with black stripes. In the GO trial, the background color in the vicinity of the fish in the virtual space changed to blue and that of the area ahead of the fish changed to red. In the NOGO trial, the color of the near-side changed to red and that of the far-side changed to blue. Success in the GO trial was defined by a correct escape to the red region by tail beats, which was initiated within 10 s after the change in the background color. Success in the NOGO trial was defined by a correct stay without tail beats in the red region for the 10 s after the change in the background color. Failure in the GO trial was a stay behavior without tail beats in the blue region after 10 s. Failure in the NOGO trial was an incorrect forward swim with tail beats into the blue region within 10 s. If the trials resulted in failure, an electric shock (5 V/cm for 1 s) was delivered from two needle electrodes placed on both sides of the body (Fig. 1b). In some fish, the color associated with the electric shock was reversed after learning with the original rule was achieved.

In the open-loop experiment, we changed the arbitrary gain constant from 10 to 0. In this situation, the fish tail beat frequency was multiplied by 0 and no feedback to the virtual reality space was generated. In the goal color change experiment, we changed the goal background color from red to green or white as soon as the fish reached the goal position.

**Two-photon imaging**. To visualize calcium signals, we used two-photon microscope LSM710-AX10 (Zeiss) with a water-immersion objective lens (W Plan-Apochromat, 20x, NA 1.0, Zeiss) and a mode-locked Ti:sapphire laser (Chameleon vision2, Coherent) at a wavelength of 890 nm. A 690 nm short-pass dichroic mirror (LP-690, Zeiss) was used to separate the excitation laser from the emitted fluorescence. Fluorescence emissions were collected using a GaAsP photomultiplier tube (BiG, Zeiss). The laser intensity was adjusted to <30 mW under the objective lens. The scanning was performed in bidirectional raster scanning with line step, 4. The imaging speed was 97.75 ms per frame. The imaged field was 384.9 × 384.9 μm (512 × 512 or 256 × 256 pixels). Imaging was performed using a custom-made piezo actuator, which allowed us to cover three planes separated by 16 μm.

**Analysis of neural activity**. Image data analysis was performed using scripts written in ImageJ (ImageJ 1.50i and 1.51p, NIH), LabVIEW (National Instruments), and Matlab (MathWorks). The imaging data derived from one slice were collected using the custom macro of ImageJ. The collected image data of each slice were processed to remove the displacement of the x–y axis during the experiment. As a reference image for such image registration, we used the average of 1000 images with minimal displacement from the dataset. We used two-dimensional Fourier transformation to determine the displacement of individual images compared to the reference image in the frequency space. In case of displacement in an image, with reference to the reference image, the distance or degree of displacement in the frequency domain was calculated in the number of pixels along the x and y axes using cross-correlation. After calculating such displacement, each image was corrected for the shift in the x and y axes. This method provided easier and more effective image registration than the TurboReg plugin module for ImageJ. After displacement correction, a median filter (radius, 2) was used to smooth the image. The images were then downsized from 512 × 512 to 256 × 256 to reduce the amount of further calculation time. In the next step, cell-like ROIs were detected from the image data. To define the ROIs corresponding to individual neurons, we used the following method: from the imaging data, the peaky-ness of all pixels, as described by Ahrens et al., 2012[27], was calculated. This peaky-ness corresponded to the change in the activity of each pixel over time. Pixels inside a cell body generally had a high peaky-ness value, as the activity of the cell caused greater deviation in the pixels inside the body than in pixels that did not belong to a cell. In the next step, we ranked the pixels by their peaky-ness values. A threshold peaky-ness value was calculated for later use by taking the average of these values. We took the pixel with the highest peaky-ness, and calculated the correlation of intensity with time between this pixel and the surrounding pixels (51 × 51 pixels). We set a threshold value of correlation empirically, and if the correlation between this pixel and one neighboring pixel was higher than the threshold, that neighboring pixel was considered to be in the same cell as this pixel. Using this process, a boundary of a cell body could be obtained. We then looked for the pixel with the second-highest peaky-ness value in the updated list, and repeated the procedure described above to identify another cell body. This process was repeated to identify cell bodies until the peaky-ness value of the pixel being considered was

reduced to less than the threshold peaky-ness value calculated earlier. After the detection of potential cells using this method, we checked the spatial overlap between the cells. If two cells overlapped, the cell with the smaller area was removed. In addition, cells detected in the area of the image where there was apparently no brain tissue were also removed after careful visual observation. After these procedures, the fluorescence timeline of each cell was obtained. For each cell, the fluorescence timelines of each pixel belonging to that cell were calculated from the image data, and an average timeline was calculated as the fluorescence of that cell. The fluorescence timelines for all cells from each slice/layer were calculated for each session of the experiment. As we recorded cellular activity from three layers, we gathered timelines of fluorescence for cells detected in all three layers. However, as each cell had only one intensity value in every three microscopic frames due to intra-layer switching, we used the spline interpolation (built-in function in LabVIEW) method to infer missing fluorescence values for each frame. With these, we could finally gather all of the cells detected over three layers with equal lengths of intensity timelines. We calculated the baseline intensity of every cell by averaging the baselines of their pixels, which were calculated by taking the average fluorescence intensity over the whole experiment. To match the behavior data to the fluorescence data, we checked the frame number of the corresponding images taken by the microscope during the experiment, and cut the time points from the fluorescence data that did not correspond to the behavior data. Finally, we calculated the $\Delta F/F_0$ for each cell using the baseline defined above.

**Template matching**. Template matching was performed as previously described[28] to evaluate whether specific neural activity patterns emerged during successful trials in both GO and NOGO trials. Briefly, this method can quantify the similarity between a neural activity pattern (template) during a certain time window and other time windows of the same length. In our case, we aimed to determine if there were specific activity patterns among neurons responsible for the successful avoidance of electric shocks in a trial, and also whether the pattern emerges during the initial phase of a trial. For the creation of templates, we considered all GO and NOGO trials after the fish successfully (success rate >80%) met the learning criterion, as the neural activity during these trials might have exhibited a specific pattern for success. We created a successful GO template using the initial 2 s of neural activity for all successful GO trials after the behavioral learning criterion was met, and taking the average over the number of such trials. We created a successful NOGO template in the same manner. Then we used the neural activity data of the entire experiment, and calculated the similarity indices between the above-mentioned templates and sliding time windows of 2 s lengths of the total experimental data. We could then derive two graphs of similarity indices, one for each template.

**Non-negative matrix factorization**. To further investigate the encoded information in neurons, we performed NMF (Fig. 3a)[20,21]. NMF separates synchronized neural activity, which enabled the analysis of the neural ensembles. The data matrix $D$ stores neural activity represented by their $\Delta F/F_0$, where each column vector stores neural activity in each time frame. NMF searches for two factor matrices ($P$ and $T$) such that matrix $D$ can be best approximated by the product of matrices $P$ and $T$ by minimizing the error function: Error $= \|D - PT\|^2$. A gradient descent method was used to minimize the error function for each given matrix $D$. In each case, there were five independent attempts with randomly assigned $P$ and $T$ matrices. The maximum number of iterations for each attempt was set to 4000. The factorization minimizing the error function was taken to be the best factorization. The column vectors of the factor matrix $P$ indicate typical neural activity patterns, i.e., neural ensembles. The corresponding row vectors of the factor matrix $T$ indicate the time course of patterns. The number of patterns, i.e., the number of columns in the left factor matrix was determined by AIC.

**Simulations**. The neural network model was formulated in the form of rate coding neurons with a sigmoid activation function. Ensembles encoding the reward prediction error $x_{RPE}$ and the scenery flow prediction error (SFPE) $x_{SFPE}$ provided modulatory signals that induced synaptic plasticity. An ensemble $x_{SF}$ received the input from a blue perceiving ensemble $x_{BP}$ and determined the probability of swimming forward. The network was modeled as $x_{SF}(t) = \text{sig}\big(Wx_{BP}(t - t_l) - h\big)$, where $t$ indicates the current time step, $W$ denotes the synaptic strength, $h$ is the fixed firing threshold, and $t_l$ is the time delay. Decision to select swimming (as opposed to stopping) at time $t$ was made by following Prob(swimming) $= x_{SF}(t - t_l)$. In the absence of the SFPE ensemble (Fig. 8d), the synaptic strength was potentiated depending only on the RPE: $\Delta W = W(t + 1) - W(t) = \eta_{RPE}x_{RPE}(t)$, which happened when the fish could avoid the punishment contrary to its expectation. Whereas, in the presence of the SFPE ensemble (Fig. 8c), $W$ was updated depending on both $x_{RPE}$ and $x_{SFPE}$: $\Delta W = \eta_{RPE}x_{RPE}(t) + \eta_{SFPE}x_{SFPE}(t)$. Note that $\eta_{RPE}$ and $\eta_{SFPE}$ denote the learning rates. Here, $x_{BP}(t) = 1$ held when the fish watched the blue color at each time step; otherwise, $x_{BP}(t) = 0$.

Moreover, the SFPE ensemble was modeled to compute the difference between the bottom-up backward flow sensation $x_{BF}$ and the top-down scenery flow prediction $x_{SFP}$, $x_{SFPE}(t) = \text{ReLU}\big(x_{SFP}(t) - x_{BF}(t)\big)$, where $\text{ReLU}(x) = \max[x, 0]$

denotes the activation function in the form of the rectified linear unit. It took a positive value when and only when the predicted scenery flow exceeds the actual sensation. Here, $x_{SFP}$ was characterized as $x_{SFP}(t) = \mathrm{sig}\big(Ux_{BP}(t) - h'\big)$ using a synaptic strength $U$. The synapse was updated by $\Delta U = \eta' x_{RPE}(t)$, through an association between sensation and punishment. Consequently, in the early stage of training, $x_{SFP}$ learned to represent $x_{SFP}(t) \approx 1$ when the fish sensed the backwardly moving scenery at each time step and otherwise $x_{SFP}(t) = 0$ (Fig. 8b). In addition, $x_{RPE}(t) = \mathrm{ReLU}\big(x_{RP} - p(t)\big)$ was determined by the gap between the actual punishment $p$ and the (negative) reward prediction $x_{RP}$. $x_{RPE}$ took a positive value when $x_{RP}$ was positive in the absence of punishment (i.e., $p = 0$); otherwise it took zero. The prediction about the punishment $x_{RP}$ was characterized as $x_{RP}(t) = \mathrm{sig}\big(Vx_{BP}(t_{init}) - h''\big)$ using a synaptic strength $V$ and the blue perception at the beginning of each trial $x_{BP}(t_{init})$. The synapse was updated by $\Delta V = -\eta'' x_{RPE}(t)$.

In the simulations, $h = 2.4$, $t_l = 10$, $\eta_{RPE} = 0.2$, $\eta_{SFPE} = 0.002$, $h' = 6$, $\eta' = 1.6$, $h'' = 4$, and $\eta'' = 1.6$ were used; $W$ and $U$ were initialized as 0; and $V$ was initialized as 8. Each trial continued 100 steps. The MATLAB code for the simulations is available upon request from the authors.

**Permutation test**. To confirm that the observed difference in the halt periods between the two groups of fish distinguished by the presence or absence of the putatively SFPE-encoding ensemble did not occur by chance, we performed a random shuffling test. For this analysis, we included four fish that possessed color rule-encoding ensembles while recording the deeper region of the dorsal telencephalon. Thus, among the total 28 fish, 19 fish expressed ensembles encoding color rules, whereas nine fish additionally possessed the putatively SFPE-encoding ensemble. The average halt periods of the 28 fish were randomly divided into two groups while the number of data in each of the two groups was fixed (group A: 9 fish data, group B: 19 fish data). After calculating the average value of halt periods in each group, we subtracted the average value of group B from that of A. We repeated this random grouping 1000 times to obtain a surrogate distribution of the differences in the halt periods, and computed the $P$-value as a fraction of the cases in which the difference became larger than the observed difference.

**The relationship between correlation coefficient in activity and distance in paired neurons**. To examine the relationship between activity similarity and the distance between neurons within ensemble, we calculated Pearson's correlation coefficient in activity between neurons and the distance. For the distance between neurons, we calculated Euclidean distance. The 10 most-contributing neurons in each ensemble were used for these calculations. The distance and correlation coefficient were plotted. For averaging trace, we divide the distance into 37.6 μm bins and averaged the correlation coefficient within each bin. For the shuffled data, we randomly chose 10 neurons in the same fish and, after shuffling their position, we calculated the correlation coefficient and the Euclidean distance. This procedure was repeated 10 times and the data were averaged.

**Identification of each ensemble**. To identify the ensembles encoding blue perception (Fig. 3), blue is dangerous (Fig. 4), red is safe (Fig. 4) and SFPE (Fig. 6), we introduced two flow charts (Supplementary Figs. 19 and 20). In these flow charts, we first selected the candidates that encode the information we focused on by observing the activities of NMF ensembles under the open-loop condition. Among the ensembles which showed increased activity under the open-loop condition, the ensembles which met the criteria in the flow chart in Supplementary Figs. 19 and 20 were regarded as blue perception, the two color rules and putative SFPE-encoding ensembles. The ensembles which did not show the specificity in the activation to each color under the closed-loop condition or the constant activation in GO or NOGO trials under the open-loop condition or only showed increased activity to red in GO or NOGO trials were abandoned. For the fish which did not experience the open-loop condition but rule reversal, we first observed the activity of ensembles after establishment of behavioral learning instead of under the open-loop condition and followed the second step in the two flow charts (Supplementary Figs. 19 and 20).

**Multiple comparison problem**. To address the problem of multiple comparison, we semi-automatized the procedure for selecting the ensembles of our interest. We first calculated the correlation coefficient between environment-dependent variables, $B(t)$ and $R(t)$, and NMF ensembles calculated from shuffled neural activity under the open-loop condition in each fish. $B(t)$ is $+1$ if fish is in the blue region, but otherwise the variable is 0. If the correlation coefficient to environment-dependent variables was higher than 0.25 (a threshold for the selection), the ensembles were regarded as candidates of the ensembles encoding the information we focused on. Among the candidates, the ensembles that met the criteria in the flow charts in Supplementary Figs. 19 and 20 were regarded as the ensemble encoding blue perception, the two color rule and putative SFPE-encoding ensembles. We first applied them to the original data from Fish 1 in Supplementary Table 1 as an example (Supplementary Fig. 16), showing the validity of these procedures for semi-automatic selection of the candidate ensembles of our interest.

By further using this method, we confirmed that the correlation coefficients rarely reached the value higher than the threshold value 0.25 by calculating correlation coefficient between NMF ensembles made from randomly shuffled neural activities derived from the Fish 1 original data and $B(t)$ or $R(t)$ (Supplementary Fig. 17). For the shuffling, two values in each neuron's activity in the time line were randomly selected and swapped, and this process was repeated $n$ times, where $n$ is the total bin number of the time line of each neuron's activity. We generated three sets of shuffled neural activity in each fish and these three shuffled neural activity sets were processed for NMF after determining the number of typical ensembles by calculating AIC in each set. After NMF calculation, we calculated the correlation coefficient between the environment-dependent variables $B(t)$ and $R(t)$ and all NMF temporal time lines in each fish.

**Quantification of activity of neural ensemble**. For quantification of the similarity to GO/NOGO templates (Fig. 2g), the peak values when fish perceived the start color in the first five trials of the adaptation, initial stage of training, and after behavioral learning was established were compared by two-tailed unpaired $t$-test. For quantification of the similarity to the GO/NOGO templates (Fig. 2g, after behavioral learning and after reaching the goal), the peak values before and after reaching the goal in the first five GO trials when fish met the behavioral learning criterion were compared by two-tailed unpaired $t$-test.

For quantification of the change in the activity of the ensemble encoding the blue perception (Fig. 3d), the peak values during GO and NOGO trials in the adaptation, in the first ten trials, which corresponded to the initial stage of training, after behavioral learning was established were compared by one-way ANOVA and Bonferroni's test. The change in the activity of the ensemble encoding the color rule that blue is dangerous (Fig. 4c, g), the peak values during GO trials in the adaptation, the initial stage of training, the stages immediately before the establishment of behavioral learning and after behavioral learning were compared by one-way ANOVA and Bonferroni's test. The change in the activity of the ensemble encoding the color rule that red is safe (Fig. 4d, h), the peak values when fish perceived red color in GO and NOGO trials in the adaptation, the initial stage of training, the intermediate stage of training, and the stages immediately before the establishment of behavioral learning and after behavioral learning were compared by one-way ANOVA and Bonferroni's test. The change in the activity of the ensembles encoding putatively the SFPE (Fig. 6f, i) was also similarly compared among the adaptation, the initial stage of training, in the successful GO trials after behavioral learning, in the failed GO trials after behavioral learning, and in the open-loop GO trials. The activity of the ensemble activated when fish perceived the goal colors that differed from red (Supplementary Fig. 14d) was also similarly compared between the states when the goal was the original red color and when the color of the goal was changed to either green or white. For the quantification in Fig. 5d, the peak values when fish perceived blue as the start color during the GO trials in the stages during adaptation and after behavioral learning under the original rule, during the NOGO trials in the initial stage of training and the stage after behavioral learning under the reversed were compared by one-way ANOVA and Bonferroni's test. For comparison, the activity normalized by self-maximum value was used.

For the quantification in Fig. 7c, total halt period, stop number and moved period in the successful GO trials when fish satisfied the learner criteria were compared by one-tailed unpaired $t$-test.

One-way ANOVA and Bonferroni's tests were performed using script written in Matlab and $t$-tests were performed using Excel (Microsoft Office 2007).

**Statistics and reproducibility**. The images shown in Fig. 2d and Supplementary Fig. 15a show averaged images of the calcium imaging data during the experiment for each fish. The averaged images of calcium imaging data of all the fish used in the experiment were similar. We randomly selected one fish's data from them and used it as the image of the figure.

**Reporting summary**. Further information on research design is available in the Nature Research Reporting Summary linked to this article.

## Data availability
The data mainly mentioned in this study (Fish 1, 2, 3 in Supplementary Table 1) are available in the repository (https://doi.org/10.5281/zenodo.5195611). The original remaining data are available from the corresponding author upon request. Source data are provided with this paper.

## Code availability
The codes used to prepare the neural activity for NMF and perform NMF and the simulation of Fig. 8 in this study are available in the repository (https://doi.org/10.5281/zenodo.5195611). The remaining custom codes are also available from the corresponding author upon request. Although the OMEGA SPACE, software used for the creation of the virtual space in this study, is not freely available, we also established the virtual space using free software whose control system is compatible with the system used for OMEGA SPACE. Although the free software was not used in this study, the system made with the free software is also available upon request.

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

## Acknowledgements

We thank Drs. Junichi Nakai, Tetsuya Koide, and Yoshihiro Yoshihara for providing G-CaMP7 plasmids, Dr. Masae Kinoshita for drawing the excellent schema of adult zebrafish, Dr. Yukiko Goda for critical reading of the manuscript, Drs. Yuki Tanimoto and Ryo Aoki for technical assistance and discussion, and Drs. Kuo-Hua Huang and Rainer W. Friedrich for communication before publication. We thank all members of the Okamoto Laboratory for support and advice, Advanced Manufacturing Support Team of RIKEN for fabricating finely tuned apparatus for fish fixation, the Research Resource Center of RIKEN Center for Brain Science for fish care, and Drs. Shinichi Higashijima and Chie Satou, and the National BioResource Project of Japan for

providing *TgBAC(vglut2a:Gal4)* fish. This work was supported by the RIKEN CBS Internal Budget, Strategic Research Program for Brain Science (H.O.) from MEXT and AMED, and Grant-in-Aid for Innovative Area (H.O., 23120008) from MEXT, the Core Research for Evolutional Science and Technology from JST and AMED (H.O., JPMJCR09S1), Grant from Kao Corporation (H.O.), Grant from Fujitsu Corporation (H.O.), the RIKEN Special Postdoctoral Researchers Program (M.T.), and Grant-in-Aid for Young Scientists (B) (M.T., 18K14858) from JSPS. T. Isomura and H.S. are supported by the grant of Joint Research by the National Institutes of Natural Sciences (NINS Program No. 01112005).

## Author contributions

H.O. conceived and supervised the project. M.T. and H.O. designed the experiments and analyses. T. Islam and M.T. built the virtual reality and computerized control and data acquisition systems. M.T. carried out the experiments. M.T., T. Isomura, H.S., and T.A. performed the analyses. T. Isomura performed the simulation. C.C.A.F. and T.F. made the NMF program. H.K. generated fish line. M.T. and H.O. wrote the manuscript with the great help of T.F., H.S., and T. Isomura.

## Competing interests

The authors declare no competing interests.
