## [Peer Review File. · Nature Communications]

Zebrafish capable of generating future state prediction error show improved active avoidance behavior in virtual realityReviewers' comments:

Reviewer #1 (Remarks to the Author):

The paper "Future state prediction error improves active avoidance behavior by adult zebrafish in virtual reality" by Torigoe et al., uses the zebrafish model to investigate neural substrates of learning. To this end, the authors develop a virtual reality system in which adult zebrafish can choose to escape from an electric shock either by swimming (a GO condition) or by staying stationary (a NOGO condition). By combining this virtual reality behavior with 2-photon calcium imaging the authors explore cell types in the forebrain of adult zebrafish that encode aspects of the learning state of the fish as well as the sensory experience in the task. The overall approach presented here, together with the zebrafish model have great promise to understand neural mechanisms underlying learned behaviors and how animals assign value to task-relevant sensory stimuli. However, there are major concerns with the current implementation of the task and the conclusions drawn. The concerns outlined below, while possible to address, would likely require a major undertaking and reframing of the paper. This is less due to the absence of critical control experiments and more due to the fact that the approach and data do not support the authors claims.

In addition, the authors do themselves a disservice, setting the reader up for disappointment, by overstating claims about the power of the model and obtained results in the introduction of the paper.

Major Concerns

1. The coverage of the telencephalon is very limited in each fish, precluding statements about the absence of cellular ensembles

The authors image 3 separate planes with an edge length of 385 μm , separated in z by 16 μm . Given the size of the adult zebrafish telencephalon (on the order of 1 mm long, 0.75 mm wide and around 200 μm thickness of the dorsal telencephalon / pallium) the approach only covers a very small fraction of telencephalic neurons. In particular, while the authors could indeed observe "almost half of a hemisphere" only a few percent are actually sampled. Therefore, it is very difficult to convincingly conclude that a given fish does not have an activity ensemble like the surprise neurons. The fact that sampling is too sparse to make this conclusion is underlined by the fact that even ensembles that are critical for the learning itself and should therefore be present in every fish (blue wall ensemble) cannot be identified in every fish.

2. It is very difficult to judge how widespread and stable ensembles are

Especially for learning related ensembles such as the ensemble that might signal that blue is dangerous it is important to demonstrate that there is consistency in responses across cells and animals. There should be a clear quantification of ensemble size, similarity of neuronal responses within ensembles and how similarly these ensembles are across fish. Without that information it is hard to judge whether the presented data truly represents an activity type or if it is just a one-off response.

3. The NOGO condition is not learned

The presentation of the NOGO condition "reaching criterion" is misleading. It seems that the default response is to stay still and that only once confusion by the paradigm sets in, the fish performs worse and then re-approaches criterion. The NOGO condition serves as an important control for some of the ensemble activities and is therefore meaningful but it should not be discussed in the context of learning.

4. The claim of an ensemble encoding the "value" of a color is problematic

The responses of the cells that are presented as encoding the value of blue when blue is paired with the shock would need further investigation. If these cells encode value it should be separated from the sensory stimulus itself. In other words, these cells should respond when blue is shown in the normal condition but then respond to red in the reversal condition. This does not seem to be the case. Instead it seems to be, that the response of these cells decreases in some instances when the stimulus pairing is reversed which then in some other instances leads the emergence of a different ensemble responding to red. It is therefore not clear how these responses can be unambiguously assigned to encode "value" rather than for example the saliency or non-saliency of

blue. Notably, the latter case would not make any reference to the meaning of the color, unlike the encoding of "value".

5. The overall stability of ensemble responses is not quantified

The interpretation of the reversal condition rests on the fact that in some fish a learning-induced blue responsive cluster disappears. However, without knowing whether ensemble responses are generally stable over longer experimental conditions, it is hard to interpret this finding. In how many fish would the same cluster disappear if the normal training condition was maintained?

6. The relationship of the SFPE ensemble and the learned behavior is intriguing but not well controlled

There are two problems relating the SFPE ensemble to the observed behavioral differences. First, given the sampling problems outlined in point 1 above, it is not clear if the fact that the sample was not found in the majority of fish can indeed be equated with those fish not having that ensemble. Second, the absence of the ensemble and the seemingly linked behavioral effect could be caused by these fish not properly seeing the moving stripes on the bottom of the tank. It is easily conceivable that a fish that is able to observe a structured floor that it virtually moves over would swim in straighter lines than a fish who cannot perceive this visual feedback. Since it is unclear how the mounting for the assay could interfere with visibility of the floor (based on the angle of the individual fish in question, positioning within the arena, etc.) it is hard to know if the absence of the ensemble even if it could be confirmed, would be causal to the different swim pattern. At least each fish later analyzed for learning should be subjected to OMR in the setup after mounting to ensure that they all react in a similar way to optic flow.

7. The computational model is overinterpreted

Based on the information present in the Main Text, Figure 7 and the Materials and Methods, the outcome of the modeling task is a foregone conclusion rather than truly testing if the SFPE ensemble is supporting learned behavior. Specifically, the model without the SFPE ensemble only learns in discrete steps, whenever a shock is followed by an escape that generates a surprise improvement in the condition. The model with the SFPE ensemble on the other hand, is updating weights whenever no optic flow is perceived in the presence of blue color ("at every time fish stops en route to the goal upon blue color presentation, fish would get the activation of this prediction error encoding ensemble [...] and it would act to strengthen the connection between the BP and SF neurons"). However, if such as learning rule existed in nature, fish should start swimming in response to blue even in the absence of an electric shock. It is therefore unclear what interpretable insight the model gives which is in large part due to the absence of biological data that could constrain it. In the very least, learning from both modules should be contingent on the escape leading to avoidance of the shock which however would be similar to just adding the two learning rates into one process.

Minor concerns

1. The paradigm is not presented well enough. It is unclear if the virtual reality moves the back wall the fish can escape to (red in the original condition) towards the fish during the escape, or if the only visual feedback is through the stripes on the bottom and the whole tank turning red after a successful escape

2. It is hard to interpret Figure 2 e-g. No units are given on the y-axis in e-f and since it is not presented what a "perfect match" and a "random match" would look like in terms of "Index to template" the plots are currently somewhat meaningless.

3. Often only some of the fish learned, performed a certain behavior or showed a certain ensemble. While not unexpected it highlights the problem with sampling sparsity and it would be helpful to see a schematic of how these different fish relate to each other. In other words, do the 2 fish that showed reversal-based suppression, have the SFPE ensemble or not, etc.

4. The text states that the number of components was chosen based on the Akaike information criterion but no plot is shown that would allow the reader to judge the quality of that choice. There should be a supplemental figure that shows AIC scores by component number.

Reviewer #2 (Remarks to the Author):

This manuscript reports on a fascinating set of discoveries about functional mechanisms of learning. The authors use adult zebrafish in a virtual reality environment while two-photon imaging many cells in their forebrains. The animals are trained in a go or nogo task to either stay in place to avoid a shock, or swim forward to a different color to avoid a shock. Some of the animals can also perform reversal learning, where the valence of the different colors of the environment are swapped. The authors use various types of analysis of the imaging data to discover a variety of functional populations of neurons in the forebrain that encode interesting quantities like associations between color and danger. These are interpreted to support a model where the animals simultaneously use two algorithms to implement the learning: reward optimization that guides the animals to safety, and surprise minimization that implicitly predicts the future state of the animal in its environment and triggers a prediction violation response in an open loop situation. What is especially interesting is that the animals with neural correlates to both models were faster at reaching the safe zone and moved in a smoother, less jerky manner. A model recapitulates the core observations. These findings have implications for theories of learning. This is a fascinating paper with multiple interesting and impactful findings.

Multiple improvements should be considered, listed below.

1) For analysis, NMF is used to find "assemblies" of cells. The technique is adequately explained in terms of spatial and temporal matrices, but it is less clear how the spatial matrices are turned into "assemblies". In NMF, each neuron gets assigned a set of weights from the spatial matrix, and activity of the neuron is approximated by the sum over time series from the temporal matrix weighted by these spatial weights. Therefore every neuron is a mix of temporal components, and conversely, every temporal component gets distributed over multiple overlapping sets of neurons. To turn these into assemblies, I assume that some kind of thresholding was applied, as suggested by supplementary figure 4b,5b,6b, and 8? Once an assembly was established, how was it analyzed - by the average activity of all neurons in the assembly?

If the temporal NMF components were directly used, the interpretation is more difficult, since it is possible that not a single neuron exists that reflects the NMF time series (since neurons are described by weighted sums over multiple time series). If the temporal components were directly used, I'd like to see single-neuron examples or averages over neurons that follows the pattern of activity described by the authors.

2) The manuscript describes results from many fish, that have been in some cases sub-selected for having populations of neurons with certain interesting activity patterns. The selection of interesting activity patterns and fish appears to have been done by eye. That is fine, but also means that a substantial amount of subjectivity in the study. Is it possible to use a more principled or automated way of deciding which activity patterns to focus on? How is the problem of multiple comparisons addressed (that if you have enough fish or neurons, even if they are random, you will find some that look interesting)?

3) The maximization of reward model, and an internal model, are not necessarily incompatible or distinct, it just so happens that these have been formulated separately. The introduction is currently phrased as if these two models are distinct (using "in contrast" and "alternative model"). I suggest changing this to reflect that it's not necessarily one or the other (as, indeed, the authors themselves conclude based on the data).

4) Line 49 "an example of the simplest behavioral paradigm", perhaps "simple" is better, since one can probably invent paradigms that are even simpler.

5) Line 59, "Its brain is very small", it depends what one compares it to, perhaps quantifying it is better?

6) Line 66, comparison to mice and monkeys. That might be true, but maybe that's because of unnatural lab conditions or so. It should be avoided to make it seem that the authors are suggesting that adult zebrafish generally learn faster than mice and monkeys.

7) The claim that only fish with the second ensemble encoding the state prediction error swim straight to the goal made me wonder, not the entire telencephalon is imaged, right? How confident are the authors that this population doesn't exist in the other fish, but was missed by the two-photon microscope?

8) The assemblies can be represented in space, as was done in the supplementary figures 4-6 and 8. It would be good to have this in the main text, also in 3d, and ideally also combined across fish, so readers can see if there are any spatial patterns where certain areas of the brain preferentially have certain neuron types.

9) Line 83, "reiterating" is not the right word here, perhaps "repeating" or "iterating" is better.

10) What is the motivation for using *camk2a* and *vglut2a* promoters?

11) The description of figure 2g is confusing to me. Why is there "after reaching goal" only shown in the "nogo template" and not in the "go template"? And are these for only the six selected fish or all of them? How are the fish selected, by eye or by some automated criterion?

12) Lines 209-214, it seems important to stress that this was not the case before learning.

13) Line 229, what does "definite" mean here, and is it the right word?

14) Figure 3D is not referred to I think.

15) The figures can be improved. I found them confusing because a lot of labels and legends are lacking, and one has to hunt in the figure legend for them. For example, in 2e, the blue and red labels "Index to..." refer to the blue and red traces, and not to the blue and red horizontal lines, which represent the stimulus. That is really confusing: please also label the blue and red horizontal lines, and explain what the higher and the lower position means (go and nogo). Please label the tail angle, the black displacement line, and the red line. What is the horizontal black line under the yellow trace?

16) Similarly it is not so clear what each dot represents for example in 2g. Is one dot a fish, or a trial? Please clarify that.

17) Line 980, $n=32$ trails or fish? Please specify both.

18) Line 1000, "color of the apposition of the fish", I think that can be easier understood if it's something like "color of the environment at the current position of the fish".

19) Figure 3c and after, here the yellow trace is labeled but it should be "bend" instead of "bent".

20) Several grammar errors need to be fixed, I can help with this in the next round.

Responses to Reviewers' comments:

(1) General Explanation on our responses to the Reviewers:

First, we would like to thank both of the reviewers for very constructive comments. We took all of the comments by the reviewers very seriously and tried to respond to all of the comments as much as possible. Throughout the new version of our manuscript, we toned down our argument by mainly describing the observed facts and removing too much conjecture on the causal relationship between prediction error and the behavior. In accordance with this change, we also changed the title of this manuscript from “**Future state prediction error improves active avoidance behavior by adult zebrafish in virtual reality**” to “**Zebrafish capable of generating future state prediction error show improved active avoidance behavior in virtual reality**”. In this new manuscript, in response to reviewers' comments, we added a series of the new data which support our argument and reveal the detailed nature of the neurons constituting a NMF ensemble. Major improvements are as the followings:

Improvement 1: In our previous manuscript, we performed three-plane imaging, raising the possibility that the reason why we could not find the putatively SFPE-encoding ensemble in some fish may be simply due to the low coverage of scanned volume of the telencephalon. In the main text, we indeed admitted that we could not perform the whole pallium imaging. However, to address the issue, we additionally performed six-plane imaging in which the coverage of the telencephalic region was increased (new Fig. 1c and new Supplementary Fig. 14). We obtained the result that the putatively SFPE-encoding ensemble was identified in the shallower region of the telencephalon (new Supplementary Table 2). This result suggests that the reason why we could not find the putatively SFPE-encoding ensemble in some fish was not simply due to the low coverage of the scanned part of the telencephalon.

Improvement 2: In our previous manuscript, we argued that fish group which has both the color rule and the putatively SFPE-encoding ensembles showed shorter halt period than the fish group which has only the color rule ensemble. To confirm our argument, we also performed the permutation test by randomly dividing all 28 fish into two groups of 9 and 19 fish. We found that the differences of halt periods between the randomly selected two groups (1000 times) rarely exceeded the observed difference between the two groups categorized by the presence or absence of a putatively SFPE-encoding ensemble (permutation P -value= 0.011 < 0.05). The result support that it was not accidental that the fish with both color rule and putatively SFPE-encoding ensembles showed a shorter halt period than the fish only with the color rule ensemble.

Improvement 3: In our previous manuscript, we did not mention about the nature of the neurons constituting a NMF ensemble. To clarify their nature, we also deepened our analysis and, as a result, we could find that the neurons with relatively highly correlated activity in the same NMF ensemble showed coincidental activation with the ensemble itself, but in an intermittent manner (new Figs. 3e, 3f and Supplementary Figs. 5-7c) and had tendency to be

localized in closer distance (new Figs. 3g, 4i and 6j and Supplementary Figs. 5-7d for individual fish), implying that the neurons in the zebrafish telencephalon encode the information in locally clustered populations.

Improvement 4: To show the robustness of our observation, we also showed two flow charts which we followed to select the neural ensembles we focused on (*i.e.* the ensembles encoding blue perception, the color rule and scenery flow prediction error (SFPE))(new Supplementary Figs. 18 and 19, shown below in Responses to Reviewer 2). The procedure is very simple and clear which may not request automatization. However, to confirm that the ensembles we focused on were not obtained by chance, we also semi-automatized this selection procedure to apply these flow charts to the data generated by random shuffling of the original neural activity. As a result, we could not find any ensemble encoding blue perception, the color rule and SFPE, suggesting that the ensembles we focused on were not obtained by chance. Altogether, these results support the robustness of our observation.

Improvement 5: In our previous manuscript, because of our insufficiency of detailed explanation, we may have caused misunderstandings on the setup of our virtual reality system by the Reviewer1. To correct such misunderstandings, we also improved our explanation on the image display system for the virtual reality experiment and added Supplementary Movie 3.

Improvement 6: While trying to respond to Reviewers' comments, we realized that the model we mentioned in the previous manuscript did not sufficiently represent the prediction error minimization. Therefore, in the new manuscript, we presented an improved computational model (new Fig. 8). The new model is not presented as the evidence for the causal involvement of the SFPE in the refinement of the behavior but as one reference for the readers to think about the mechanistic explanation of our observation. Besides, this new model addressed the Reviewer 1's concern that the SFPE could emerge independently from punishment (electric shock). In the new model, the SFPE emerges only by the punishment but not without it (new Fig. 8a).

Improvement 7: To respond to the request by the Reviewer 2, we added the spatial distribution of neurons which contribute to the ensembles encoding blue perception, color rules and SFPE in all fish in new Supplementary Figs. 8, 9, 10, 12. Although we could not find the similarity of their distribution in the brain region in all types of ensembles, we found as shown in **Improvement 3** that the neurons with relatively highly correlated activity in the same NMF ensemble had the tendency to be localized in closer distance.

Improvement 8: To address the comments by the Reviewer 1 whether the diminishment of the ensemble is caused by rule change or by the instability of the ensemble, we studied the group of fish which experienced only the original rule but performed trials for a longer period than the group of fish which experienced reversal of the rule. In the former group, the ensemble that showed increased activity upon presentation of blue color in repeated trials continued to be

activated until the end of the experiment period. The continued activation contradicts the possibility that instability of the ensemble might cause the diminished activity of the ensemble encoding the rule that blue is dangerous after rule reversal.

Improvement 9: In introduction, we changed the statement about the relationship between reward maximization theory and surprise minimization theory from the mutually exclusive relationship to the compatible relationship according to Reviewer 2's comment.

Improvement 10: In Discussion, we added new discussion about a possible reason why the ensemble encoding blue perception was not observed in all fish in response to the Reviewer 1's comment.

Altogether, I hope that these improvements have made our new manuscript worth publication in your journal. We will explain our point-by-point responses to the comments in the following pages.

(2) Point-by-Point Responses to Reviewer #1:

First of all, we would like to express our deepest thanks to the Reviewer 1 for many constructive comments.

Throughout the new version of our manuscript, we toned down our argument by mainly describing the observed facts and removing too much conjecture on the causal relationship between prediction error and the behavior especially in Results. In accordance with this change, we also changed the title of this manuscript from “Future state prediction error improves active avoidance behavior by adult zebrafish in virtual reality” to “Zebrafish capable of generating future state prediction error show improved active avoidance behavior in virtual reality”.

For this new version of the manuscript, we additionally performed six-plane imaging to increase the coverage of the telencephalic region. As a result, although we also admit (explicitly in the new main text) that we could not perform the whole pallium imaging, we obtained the result suggesting that the reason why we could not find the SFPE-encoding ensemble in some fish may not be simply due to the low coverage of scanned part of the telencephalon. We also deepened our analysis about the nature of the neurons constituting a NMF ensemble and, as a result, we could find that the neurons with highly correlated activity in the same NMF ensemble have tendency to be localized in closer distance, implying that the neurons in the zebrafish telencephalon encode the information in locally clustered populations.

We also improved our explanation on the image display system for the virtual reality experiment to correct misunderstanding on this system by the Reviewer 1 due to our insufficiency of detailed explanation. In fact, the scenery shown by all four displays on the left, right, front and bottom coordinately shifted backwardly in response to the tail beating by fish in the closed-loop condition.

In the new manuscript, we presented the improved computational model not as an evidence for the causal involvement of the SFPE in the refinement of the behavior but as one reference for the readers to think about the mechanistic explanation of our observation.

We will explain our point-by-point responses to the comments below.

The paper “Future state prediction error improves active avoidance behavior by adult zebrafish in virtual reality” by Torigoe et al., uses the zebrafish model to investigate

neural substrates of learning. To this end, the authors develop a virtual reality system in which adult zebrafish can choose to escape from an electric shock either by swimming (a GO condition) or by staying stationary (a NOGO condition). By combining this virtual reality behavior with 2-photon calcium imaging the authors explore cell types in the forebrain of adult zebrafish that encode aspects of the learning state of the fish as well as the sensory experience in the task. The overall approach presented here, together with the zebrafish model have great promise to understand neural mechanisms underlying learned behaviors and how animals assign value to task-relevant sensory stimuli. However, there are major concerns with the current implementation of the task and the conclusions drawn. The concerns outlined below, while possible to address, would likely require a major undertaking and reframing of the paper. This is less due to the absence of critical control experiments and more due to the fact that the approach and data do not support the authors claims.

In addition, the authors do themselves a disservice, setting the reader up for disappointment, by overstating claims about the power of the model and obtained results in the introduction of the paper.

Major Concerns

1. The coverage of the telencephalon is very limited in each fish, precluding statements about the absence of cellular ensembles

The authors image 3 separate planes with an edge length of 385 μm , separated in z by 16 μm . Given the size of the adult zebrafish telencephalon (on the order of 1 mm long, 0.75 mm wide and around 200 μm thickness of the dorsal telencephalon / pallium) the approach only covers a very small fraction of telencephalic neurons.

Response: In new Figs. 1c, d, we show the volume (area and thickness) of the telencephalon which we imaged. Actually we also tried to perform the 9 plane imaging to cover the major part of the dorsal pallium (16 μm step, covering 8X16=128 μm), but the fluorescence signal derived from the deepest brain region became very low and the increase of laser power to capture the signal derived from the deepest brain region was not suitable for longer imaging especially for surface planes (around 90min in this study). Therefore, we were obliged to give up imaging the whole pallium. Therefore, we rewrote the manuscript as the following. In the manuscript, we newly described that we focused

in this study on the surface part of Dc as a first step to study the entire part of telencephalon in the future (Line 119-130).

“The zebrafish telencephalon is reported to contain the homologous regions of the isocortex (central zone of the dorsal pallium, Dc), hippocampus (lateral zone of the dorsal pallium, Dl) and amygdala (medial zone of the dorsal pallium, Dm²⁶; Fig. 1c). The best way to reveal the role of these regions in decision-making would be to capture neural activity from all of these regions. However, due to various technical limitations of conventional 2-photon calcium imaging at present, we focused our observation of neural activity imaging mainly on the surface part of Dc, the putative zebrafish homolog of the mammalian isocortex, because the isocortex plays an important role in mammalian decision-making. (Figs. 1c, d). The area we observed occupied the large surface part of the Dc and a part of the other dorsal pallial regions, such as Dm and Dl (Figs. 1c, d). To achieve the extensive labeling of the excitatory neurons, we used a triple transgenic line for *camk2a: gal4*, *vglut2a: gal4* and *UAS: G-CaMP7* (Fig. 1d).”

In particular, while the authors could indeed observe “almost half of a hemisphere” only a few percent are actually sampled. Therefore, it is very difficult to convincingly conclude that a given fish does not have an activity ensemble like the surprise neurons. The fact that sampling is too sparse to make this conclusion is underlined by the fact that even ensembles that are critical for the learning itself and should therefore be present in every fish (blue wall ensemble) cannot be identified in every fish.

Response: To address the first point that the reason why some fish did not have the putative scenery flow prediction error (SFPE) ensemble could be due to the low coverage of the telencephalon, we newly extended imaging into deeper telencephalic region by increasing the number of imaging planes from three to six (new Fig. 1c, red-boxed area and new Supplementary Fig. 14a which showed the average image of each plane in six-plane calcium imaging). We newly described the results in the text as follows (Line 484-499). From the result, although we could not perform the whole pallium imaging, it is reasonable to think that the possibility that we missed the SFPE-encoding ensemble due to the low coverage rate might not be so high.

“The putatively SFPE-encoding ensemble was observed in eight out of 24 fish that possessed a color rule-encoding ensemble. We examined the possibility that the failure to detect the putatively SFPE-encoding ensemble in the remaining 16 fish might result from the limitation in the scanning volume of the telencephalon. To address this question, we extended

imaging into a deeper telencephalic region by increasing the number of imaging planes from three to six (Supplementary Fig. 14a shows the averaged calcium images of six planes). NMF calculations were performed in the deeper three planes and the superficial three planes separately and compared. In one out of four learner fish, we identified the putatively SFPE-encoding ensemble. Consistent with the previous results, this fish showed shorter halt periods than the fish with only the color rule-encoding ensemble (Supplementary Fig. 14b). Supplementary Fig. 14b shows the halt period of the two groups. The putatively SFPE-encoding ensemble was observed in the surface planes but not in the deeper planes (Supplementary Table 2 summarizes results of identified ensembles in surface and deeper planes in the four fish), implying that the surface three-plane imaging might be enough to capture the putatively SFPE-encoding ensemble.”

Further, to exclude the possibility that the difference in the halt time between the two groups of fish was accidentally observed, we performed the data random shuffling as we newly described in Line 500-509 as the following.

“The permutation test of randomly dividing all 28 fish into two groups of 9 and 19 fish revealed that the difference of halt periods between two groups randomly made for 1000 times rarely exceeded the actually observed difference between the two groups categorized by the presence or absence of a putatively SFPE-encoding ensemble (permutation P -value= 0.011 < 0.05), confirming that the shorter halt period of fish with both color rule and putatively SFPE-encoding ensembles was not accidental.

Therefore, this statistical analysis confirmed that the fish with both ensembles swam forward more efficiently, *i.e.* swam straight toward the goal, than the fish with only the ensemble assigning a ‘blue-dangerous’ rule, which paused for a longer period on the way to the goal (Fig. 7a, b and Supplementary PowerPoint file).”

To address the 2nd issue that the blue perception ensemble was not observed in all fish, we also examined whether blue perception-encoding ensemble was observed in the new six-plane imaging. In one out of four fish, we did not observe the blue perception-encoding ensemble (new Supplementary Table 2). Based on this result, we added in Discussion the following sentence (Line 670-677).

“Although we observed neural activity in a relatively wide region of the dorsal telencephalon, we could not cover the entire dorsal telencephalon. In fact, although

electrophysiology studies in percomorphs revealed the visual sensory input to the dorsal part of the DI region from the preglomerular nucleus, suggesting the equivalence of this region to the primary visual cortex in mammals⁴⁴, we could not fully image this area in this study. This may be why we missed the ensemble encoding blue perception in some fish. Further experiments for capturing neural activity in the wider and thicker areas of the telencephalon are necessary to address this limitation.”

2. It is very difficult to judge how widespread and stable ensembles are.

Especially for learning related ensembles such as the ensemble that might signal that blue is dangerous it is important to demonstrate that there is consistency in responses across cells and animals. There should be a clear quantification of ensemble size, similarity of neuronal responses within ensembles and how similarly these ensembles are across fish. Without that information it is hard to judge whether the presented data truly represents an activity type or if it is just a one-off response.

Response: To respond to the request for the quantification of ensemble size, we newly showed the number of neurons within ensemble in new Supplementary Figs. 5-7b.

To analyze the similarity of neuronal activities within the same ensemble, we newly added the analysis on the neurons in the blue perception ensemble as described in Line 243-259 as the following.

“To further reveal details of this ensemble encoding the perception of blue color, we plotted the contribution of each neuron within the ensemble (Fig. 3e, upper panel and Supplementary Figs. 5-7b, middle panel in left column, blue perception) and the correlation coefficient of each neuron’s activity to the ensemble’s activity (Fig. 3e, bottom panel and Supplementary Figs. 5-7b, bottom panel in left column, blue perception). Supplementary Figs. 5-7b show the spatial distributions of various neural ensembles (b, upper panels), the contribution of neurons within the ensemble (b, middle panels) and the correlation coefficient of each neuron’s activity to the activity (b, bottom panels) of each ensemble. Although one neuron had by far the highest level of contribution, other neurons with relatively lower contributions also showed significantly high levels of positive correlation. A further comparison between the ensemble activity and each neuron’s activity revealed that the neurons with lower levels of contribution but with relatively high correlation coefficients showed coincidental activation with the ensemble, but in an intermittent manner (Fig. 3f and Supplementary Figs. 5-7c, upper

left panel, blue perception). Supplementary Figs. 5-7c show the activities of the ensemble (top trace) and the five most-contributing neurons in the ensemble (descending order from the top).”

We obtained the similar results for the neurons in the other ensembles as we newly described for the neurons in the color rule-encoding neurons (Line 314-320) and in the putatively SFPE encoding ensemble (Line 447-453). The following sentences were added to the text.

(Line 314-320): “We observed a similar tendency in the relationships of the activity of individual neurons within an ensemble to the activity of the whole ensemble as in the blue perception coding ensemble. Namely, even neurons with lower levels of contribution showed relatively high correlation coefficients to the ensemble activity with intermittent coincidental activation with the ensemble (Supplementary Figs. 5-7b, 2nd and 3rd columns from the left show the results of the ensembles encoding the rule that blue is dangerous and the rule that red safe, respectively).”

(Line 447-453): “These ensembles also showed a similar tendency in the relationships of the activities of individual neurons within an ensemble to the activity of the ensemble itself as we observed in the blue perception coding ensemble. Namely, even the neurons with lower levels of contribution showed relatively high correlation coefficients to the ensemble activity with intermittent coincidental activation with the ensemble (Supplementary Figs. 5-7b, right column and c, bottom right panel, scenery flow prediction error).”

About the similarity of spatial distribution of each ensemble across fish, we could not identify the similarity of the ensemble in the distribution of the brain region in all types of ensembles (new Supplementary Fig. 8 for blue perception, 9 for the rule that blue is dangerous, 10 for the rule that red is safe and 12 for SFPE). However, as shown in new Figs. 3g, 4i, 6j and new Supplementary Figs. 5-7d for individual fish, we newly examined the correlation coefficient between 10 most-contributing neurons and their distance in all types of ensembles as described as the following (For blue perception, Line 260-275; for color rules, Line 320-326; for SFPE, Line 453-459).

(Line 260-275): “To evaluate the distribution of neurons within the ensemble, we plotted the neurons with their weight of contribution in the ensemble, but we did not identify the brain region where the neurons within the blue perception ensemble preferentially accumulated.

Supplementary Fig. 8 shows the distribution of the neurons within the blue perception ensemble across different fish.

However, to further evaluate the spatial distribution of the neurons within the ensemble, we calculated the correlation coefficient of the ten neurons with the highest levels of contribution to the ensemble. We then plotted the relationship between the paired correlation coefficients and the distance of these neurons after averaging the data derived from all 27 fish that had the ensemble encoding blue color perception (Fig. 3g; for individual fish, please see Supplementary Figs. 5-7d, upper left panels, blue perception). The result showed that neurons with highly correlated activity have a tendency to be localized at a closer distance, implying that neurons in the zebrafish telencephalon encode the information in locally clustered populations. Supplementary Figs. 5-7d show the correlation between the correlation coefficients of paired neurons' activities and their distance in each ensemble for individual fish."

(Line 320-326): "These neurons tended to be localized at a closer distance with highly correlated activity (Fig. 4i; for individual fish, please see Supplementary Figs. 5-7d), although these ensembles did not have the particular brain region where the neurons within the ensembles of the same nature accumulated across different fish. Supplementary Figs. 9 and 10 show the distribution of the neurons within the two-color rule-encoding ensembles, *i.e.* blue is dangerous and red is safe."

(Line 453-459): "These neurons tended to be localized at a closer distance with highly correlated activity (Fig. 6j; for individual fish, please see Supplementary Figs. 5-7d, bottom right panel, scenery flow prediction error) although these ensembles did not have the particular brain region where the neurons within the putative SFPE-encoding ensemble accumulated across different fish. Supplementary Fig. 12 shows the distribution of the neurons within the putative SFPE-encoding ensemble across different fish."

3. The NOGO condition is not learned

The presentation of the NOGO condition "reaching criterion" is misleading. It seems that the default response is to stay still and that only once confusion by the paradigm sets in, the fish performs worse and then re-approaches criterion. The NOGO condition serves as an important control for some of the ensemble activities and is therefore meaningful but it should not be discussed in the context of learning.

Response: Actually, we admit that fish tended to stay in VR tank. Therefore, we newly described the following sentence in the text (Line 160-164) and removed “or stay” in subheading (Line 132).

“In contrast, because of the fish’s tendency to stop tail beating under the tethered condition in the VR arena, we could not conclude that fish actually learned to stay in the red region in NOGO task only from these behavioral data. However, we detected certain change in the neural activities following repeated NOGO trials as described below.”

In contrast, we detected certain change in the neural activities following repeated NOGO trials in the template matching analysis and NMF analysis. In particular, we found the ensemble which is likely to encode the rule that red is safe by rule reversal. Therefore, we newly described the result in the text (Line 378-385) as shown below and added new Supplementary Fig. 11, magenta line.

“In the two fish that experienced rule reversal, we also observed the ensemble that got repeatedly activated upon perception of red under the original rule. The activity of this ensemble diminished after rule reversal (Supplementary Fig. 11, magenta line under both original and reversed rules), further supporting our idea that the ensemble encodes the rule that red is safe. Although we could not conclude from the behavioral data that fish learn the stop behavior in the NOGO task under the original rule, the generation of the ensemble that is likely to encode the rule that red is safe implies that fish might have learned to recognize safety when fish perceived red color.”

4. The claim of an ensemble encoding the “value” of a color is problematic

The responses of the cells that are presented as encoding the value of blue when blue is paired with the shock would need further investigation. If these cells encode value it should be separated from the sensory stimulus itself. In other words, these cells should respond when blue is shown in the normal condition but then respond to red in the reversal condition. This does not seem to be the case. Instead it seems to be, that the response of these cells decreases in some instances when the stimulus pairing is reversed which then in some other instances leads the emergence of a different ensemble responding to red. It is therefore not clear how these responses can be unambiguously assigned to encode “value” rather than for example the saliency or

non-saliency of blue. Notably, the latter case would not make any reference to the meaning of the color, unlike the encoding of “value”.

Response: In this study, we would like to argue that the ensemble encodes the context dependent rule which cannot be quantified by the absolute scalar value. Therefore, we changed the term “value” to “rule” in the text except in the citation of other paper.

5. The overall stability of ensemble responses is not quantified

The interpretation of the reversal condition rests on the fact that in some fish a learning-induced blue responsive cluster disappears. However, without knowing whether ensemble responses are generally stable over longer experimental conditions, it is hard to interpret this finding. In how many fish would the same cluster disappear if the normal training condition was maintained?

Responses: To segregate the possibilities that the diminishment of the ensemble is caused by rule change or by the instability of the ensemble, we newly examined the activity of the ensemble encoding the rule that blue is dangerous in two groups and, based on the result, we added the following sentence (Line 363-371).

“Another group of fish (eight out of 32 fish) experienced only the original rule but performed trials for a longer period than the group of fish that experienced both rules. The averaged ensemble activities in the last 10 GO trials (L) in these eight fish were not below the averaged ensemble activities immediately after the establishment of behavioral learning (I) ($L/I=3.25\pm 0.97$), indicating that the ensemble that showed increased activity upon presentation of blue color in repeated trials continued to be responsive until the end of the experiment period. The continued activation contradicts the possibility that habituation might cause the diminished activity of the ensemble encoding the rule that blue is dangerous after rule reversal.”

6. The relationship of the SFPE ensemble and the learned behavior is intriguing but not well controlled

There are two problems relating the SFPE ensemble to the observed behavioral differences. First, given the sampling problems outlined in point 1 above, it is not clear if the fact that the sample was not found in the majority of fish can indeed be equated with those fish not having that ensemble. Second, the absence of the ensemble and the seemingly linked behavioral effect could be caused by these fish not properly seeing the moving stripes on the bottom of the tank. It is easily conceivable that a fish that is able

to observe a structured floor that it virtually moves over would swim in straighter lines than a fish who cannot perceive this visual feedback. Since it is unclear how the mounting for the assay could interfere with visibility of the floor (based on the angle of the individual fish in question, positioning within the arena, etc.) it is hard to know if the absence of the ensemble even if it could be confirmed, would be causal to the different swim pattern. At least each fish later analyzed for learning should be subjected to OMR in the setup after mounting to ensure that they all react in a similar way to optic flow.

Response: About the 1st point that absence of SFPE in the majority of fish is due to the low coverage of the telencephalon, the answer is mentioned in the point 1 above.

About the 2nd point that the absence of the ensemble and the seemingly linked behavioral effect could be caused by these fish not properly seeing the moving stripes on the bottom of the tank, the gray arrow under VR tank in previous Fig. 1a might have caused the misunderstanding by the Reviewer 1. Actually, the VR scenery presented on all displays in all directions moved backwardly together according to the tail beat. In addition to this, the bottom of fixation apparatus is transparent. Therefore, fish can perceive the movement of scenery in all directions including the bottom scenery. Therefore, we think that OMR is not necessary for this experiment. To avoid this misunderstanding, we added Supplementary Movie 3 which showed the actual tail beat and the feedback scenery movement in the VR presented on all four displays. We also changed annotation in new Fig.1a from a simple gray arrow to the verbal explanation “in response to the tail beat, all landscapes moved backward” and added the statement about the feedback scenery movement in the text as the following (Line 110-114).

“As the fish beat their tail, the visual images of the scenery presented on all the surrounding displays on the left, right, front, and bottom sides of the tank were shifted backward according to the calculated virtual traveling distance of the fish to make the system closed-loop (Fig. 1a and Methods). The actual tail behavior and the feedback scenery movement are shown in Supplementary Movie 3.”

7. The computational model is overinterpreted

Based on the information present in the Main Text, Figure 7 and the Materials and Methods, the outcome of the modeling task is a foregone conclusion rather than truly testing if the SFPE ensemble is supporting learned behavior. Specifically, the model

without the SFPE ensemble only learns in discrete steps, whenever a shock is followed by an escape that generates a surprise improvement in the condition. The model with the SFPE ensemble on the other hand, is updating weights whenever no optic flow is perceived in the presence of blue color (“at every time fish stops en route to the goal upon blue color presentation, fish would get the activation of this prediction error encoding ensemble [...] and it would act to strengthen the connection between the BP and SF neurons”). However, if such a learning rule existed in nature, fish should start swimming in response to blue even in the absence of an electric shock. It is therefore unclear what interpretable insight the model gives which is in large part due to the absence of biological data that could constrain it. In the very least, learning from both modules should be contingent on the escape leading to avoidance of the shock which however would be similar to just adding the two learning rates into one process.

Response: In the new manuscript, we proposed a new model because we realized that the model we mentioned in the previous manuscript did not suitably represent the prediction error minimization. This was not presented as the evidence for the causal involvement of the SFPE in the refinement of the behavior, but as one reference to help the readers to think about the underlying mechanism.

In our new model, some of the Reviewer 1’s concern is addressed. The SFPE emerged only by the punishment (electric shock) but not in nature (new Fig. 8a). We newly explained the model by adding the following sentences in the text (Line 512-560). In addition to this, we tone-downed our argument by changing the headline of the section to “Computational models recapitulate behavioral difference” (Line 511 subheading).

“Based on the aforementioned empirical observations, we developed a neural network model of fish. We supposed that the input from the blue perception neurons (BP neurons; Fig. 8a, blue circle) activated downstream neurons that generate motor signals to make fish swim forward (SF neurons; Fig. 8a, red circle). The strength of the input was updated through an activity-dependent plasticity regulated by signals of two ensembles encoding the reward prediction error (RPE) and SFPE (Fig. 8a, green and cyan circles). We further supposed that the SFPE ensemble computed the difference between the bottom-up backward flow signals and the top-down backward scenery flow prediction (SFP); whereas the SFP ensemble was self-organized through plasticity mediated by the RPE ensemble. This model could simulate the emergence of the SFP and SFPE ensembles in the network through an association between

visual perception and punishment—consistent with our empirical observations of the emergence of the SFPE ensemble in the training stage (Fig. 8b; compare with Fig. 6d).

Using this computational model, we examined whether our observation—*i.e.* the straight swimming pattern of the fish with the SFPE ensemble—could be explained by the hypothesis that the SFPE played a role in facilitating taking optimal action to minimize the prediction error between the real observed scenery and the predicted backward moving scenery.

At each time point on its way to the goal during blue color presentation, we can assume that the fish makes a decision whether to go forward or stop depending on how strongly the input from the BP neurons can activate downstream neurons that make fish swim forward (*i.e.* SF neurons). If the aforementioned hypothesis is true, every time the fish stops *en route* to the goal during blue color presentation and the backward flow of the scenery is interrupted, the fish would experience the activation of the SFPE ensemble. The SFP ensemble would expect its presence, inducing the SFPE signal. Here, the SFPE signal would act to strengthen the connection between the BP and SF neurons. This would hinder the fish from stopping and favor a GO choice with a gradually higher probability, because the SFPE can be minimized by this behavioral choice. After repetition of this process, the fish would eventually learn to swim straight to the safe goal without choosing to stop at any point on its way to the goal. This swimming pattern is consistent with our observations, and the simulated fish model could replicate empirically observed fish behaviors (Fig. 7a, 8a, 8c). These results support the role of the SFPE ensemble in the strengthening of the connectivity between BP and SF.

In contrast, the fish with only the ensemble assigning rules to colors learn the escape behavior according to the naïve reinforcement learning, as we described previously⁹. Here, receiving punishment by staying in the blue region brings about an expected reward level, and avoiding punishment by moving into the red region generates a positive RPE. In this case, the connectivity between the BP and SF neurons is strengthened relatively infrequently, *i.e.* only when the fish happens to successfully reach the safe red goal. Moreover, the strengthening of this connectivity ceases at a relatively early stage of the training; as soon as it reaches a level enough to bring the fish to the goal within the given time limit before shock is given. With this still relatively low level of connectivity, the activation of the BP neurons cannot always activate the SF neurons. Thus, the fish would keep the tendency to stop on its way to the goal, even after they become able to constantly reach the safe goal (Fig. 7b, 8a, 8d). Hence, we could attribute the mechanisms underlying two different swimming strategies we observed to the facilitation of the connectivity between the BP and SF neurons induced by the SFPE ensemble.”

Minor concerns

1. The paradigm is not presented well enough. It is unclear if the virtual reality moves the back wall the fish can escape to (red in the original condition) towards the fish during the escape, or if the only visual feedback is through the stripes on the bottom and the whole tank turning red after a successful escape

Response: The response to this comment is the same as the 2nd answer in the point 6 of the major concern about "The relationship of the SFPE ensemble and the learned behavior is intriguing but not well controlled" above.

2. It is hard to interpret Figure 2 e-g. No units are given on the y-axis in e-f and since it is not presented what a "perfect match" and a "random match" would look like in terms of "Index to template" the plots are currently somewhat meaningless.

Response : To make the interpretation easy, we added the labels for GO, NOGO period, tail bend angle, traveled distance (the black displacement line), goal border (the red line) and 0 for index to templates to new Fig. 2e. In the text, we mentioned that "The value of the similarity index varied from -1 (anti-correlation) to $+1$ (perfect correlation). 0 of this index means no correlation." (Line 188-190).

3. Often only some of the fish learned, performed a certain behavior or showed a certain ensemble. While not unexpected it highlights the problem with sampling sparsity and it would be helpful to see a schematic of how these different fish relate to each other. In other words, do the 2 fish that showed reversal-based suppression, have the SFPE ensemble or not, etc.

Response: We put Supplementary Table1 which indicate the presence or absence of each ensemble in all learner fish. Actually, the fish which experienced rule reversal did not experience open-loop. Therefore, to identify the putatively SFPE-encoding ensemble in these fish, we needed to judge only from the activity difference between failure GO and success GO trials. As a result, the two fish which showed reversal-based suppression did not have the putatively SFPE-encoding ensemble.

Although we did not identify the putatively SFPE-encoding ensemble in these two fish, which showed reversal-based suppression of the activity of ensemble encoding the rule that blue is dangerous, we observed that the activity of the ensemble

which is regarded as encoding blue perception in the original rule continued to be reactive to blue scenery even in the reversed rule. We newly mention this in the text (Line 372-377).

“In these four fish that experienced rule reversal, we observed the other ensemble that showed increased activity whenever fish perceived blue color in a manner independent of behavioral learning. This ensemble continued to be activated whenever fish saw blue color, both in the GO trials under the original rule, and in the NOGO trials after rule reversal (Supplementary Fig. 11, blue line), supporting that this ensemble encoding the perception of blue color was not subject to rule reversal.”

4. The text states that the number of components was chosen based on the Akaike information criterion but no plot is shown that would allow the reader to judge the quality of that choice. There should be a supplemental figure that shows AIC scores by component number.

Response: We added new Supplementary Fig. 3 which indicates the AIC curve in all learner fish in the original rule. Red arrow and dot in the figure indicate the minimum number of AIC.

(3) Point-by-Point Responses to Reviewer #2:

First of all, we would like to express our deepest thanks to the Reviewer 2 for many constructive comments. Although we did our best to improve English writing of the manuscript by receiving help from the native speakers, we would still greatly appreciate if you could generously afford to help us for further improvements as you mentioned in your previous comments.

Throughout the new version of our manuscript, we toned down our argument by mainly describing the observed facts and removing too much conjecture on the causal relationship between prediction error and the behavior. In accordance with this change, we also changed the title of this manuscript from “Future state prediction error improves active avoidance behavior by adult zebrafish in virtual reality” to “Zebrafish capable of generating future state prediction error show improved active avoidance behavior in virtual reality”.

For this new version of the manuscript, we additionally performed six-plane imaging to increase the coverage of the telencephalic region. As a result, although we also admit that we could not perform the whole pallium imaging, we obtained the result suggesting that the reason why we could not find the SFPE-encoding ensemble in some fish may not be simply due to the low coverage of scanned part of the telencephalon. We also deepened our analysis about the nature of the neurons constituting a NMF ensemble and, as a result, we could find that the neurons with highly correlated activity in the same NMF ensemble have tendency to be localized in closer distance, implying that the neurons in the zebrafish telencephalon encode the information in locally clustered populations.

We also improved our explanation on the image display system for the virtual reality experiment to correct misunderstanding on this system by the Reviewer 1 due to our insufficiency of detailed explanation.

In the new manuscript, we presented the improved computational model not as the evidence for the causal involvement of the SFPE in the refinement of the behavior but as one reference for the readers to think about the mechanistic explanation of our observation.

We will explain our point-by-point responses to the comments below.

This manuscript reports on a fascinating set of discoveries about functional mechanisms of learning. The authors use adult zebrafish in a virtual reality environment

while two-photon imaging many cells in their forebrains. The animals are trained in a go or nogo task to either stay in place to avoid a shock, or swim forward to a different color to avoid a shock. Some of the animals can also perform reversal learning, where the valence of the different colors of the environment are swapped. The authors use various types of analysis of the imaging data to discover a variety of functional populations of neurons in the forebrain that encode interesting quantities like associations between color and danger. These are interpreted to support a model where the animals simultaneously use two algorithms to implement the learning: reward optimization that guides the animals to safety, and surprise minimization that implicitly predicts the future state of the animal in its environment and triggers a prediction violation response in an open loop situation. What is especially interesting is that the animals with neural correlates to both models were faster at reaching the safe zone and moved in a smoother, less jerky manner. A model recapitulates the core observations. These findings have implications for theories of learning. This is a fascinating paper with multiple interesting and impactful findings.

Multiple improvements should be considered, listed below.

1) For analysis, NMF is used to find "assemblies" of cells. The technique is adequately explained in terms of spatial and temporal matrices, but it is less clear how the spatial matrices are turned into "assemblies". In NMF, each neuron gets assigned a set of weights from the spatial matrix, and activity of the neuron is approximated by the sum over time series from the temporal matrix weighted by these spatial weights. Therefore every neuron is a mix of temporal components, and conversely, every temporal component gets distributed over multiple overlapping sets of neurons. To turn these into assemblies, I assume that some kind of thresholding was applied, as suggested by supplementary figure 4b,5b,6b, and 8? Once an assembly was established, how was it analyzed - by the average activity of all neurons in the assembly?

If the temporal NMF components were directly used, the interpretation is more difficult, since it is possible that not a single neuron exists that reflects the NMF time series (since neurons are described by weighted sums over multiple time series). If the temporal components were directly used, I'd like to see single-neuron examples or averages over neurons that follows the pattern of activity described by the authors.

Response: Actually, we directly used the temporal NMF components. In the previous version of our manuscript, in Supplementary Figs. 4-6c, we put the averaged neural activity in each type of trials, GO trials in the adaptation etc.

In the new version of our manuscript, we newly examined the similarity of ensemble's activity and neuronal activities within the same ensemble. We newly added the analysis on the neurons in the blue perception ensemble as described in Line 243-259 as the following.

“To further reveal details of this ensemble encoding the perception of blue color, we plotted the contribution of each neuron within the ensemble (Fig. 3e, upper panel and Supplementary Figs. 5-7b, middle panel in left column, blue perception) and the correlation coefficient of each neuron's activity to the ensemble's activity (Fig. 3e, bottom panel and Supplementary Figs. 5-7b, bottom panel in left column, blue perception). Supplementary Figs. 5-7b show the spatial distributions of various neural ensembles (b, upper panels), the contribution of neurons within the ensemble (b, middle panels) and the correlation coefficient of each neuron's activity to the activity (b, bottom panels) of each ensemble. Although one neuron had by far the highest level of contribution, other neurons with relatively lower contributions also showed significantly high levels of positive correlation. A further comparison between the ensemble activity and each neuron's activity revealed that the neurons with lower levels of contribution but with relatively high correlation coefficients showed coincidental activation with the ensemble, but in an intermittent manner (Fig. 3f and Supplementary Figs. 5-7c, upper left panel, blue perception). Supplementary Figs. 5-7c show the activities of the ensemble (top trace) and the five most-contributing neurons in the ensemble (descending order from the top). “

We obtained the similar results for the neurons in the other ensembles as we newly described for the neurons in the color rule-encoding neurons (Line 314-320) and in the putatively SFPE encoding ensemble (Line 447-453). The following sentences were added to the text.

(Line 314-320): “We observed a similar tendency in the relationships of the activity of individual neurons within an ensemble to the activity of the whole ensemble as in the blue perception coding ensemble. Namely, even neurons with lower levels of contribution showed relatively high correlation coefficients to the ensemble activity with intermittent coincidental activation with the ensemble (Supplementary Figs. 5-7b, 2nd and 3rd columns from the left

show the results of the ensembles encoding the rule that blue is dangerous and the rule that red safe, respectively).”

(Line 447-453): “These ensembles also showed a similar tendency in the relationships of the activities of individual neurons within an ensemble to the activity of the ensemble itself as we observed in the blue perception coding ensemble. Namely, even the neurons with lower levels of contribution showed relatively high correlation coefficients to the ensemble activity with intermittent coincidental activation with the ensemble (Supplementary Figs. 5-7b, right column and c, bottom right panel, scenery flow prediction error).”

Furthermore, as shown in new Figs. 3g, 4i, 6j and new Supplementary Figs. 5-7d for individual fish, we newly examined the correlation coefficient between 10 most-contributing neurons and their distance in all types of ensembles as described as the following (For blue perception, Line 265-275; for color rules, Line 320-326; for SFPE, Line 453-459).

(Line 265-275): “However, to further evaluate the spatial distribution of the neurons within the ensemble, we calculated the correlation coefficient of the ten neurons with the highest levels of contribution to the ensemble. We then plotted the relationship between the paired correlation coefficients and the distance of these neurons after averaging the data derived from all 27 fish that had the ensemble encoding blue color perception (Fig. 3g; for individual fish, please see Supplementary Figs. 5-7d, upper left panels, blue perception). The result showed that neurons with highly correlated activity have a tendency to be localized at a closer distance, implying that neurons in the zebrafish telencephalon encode the information in locally clustered populations. Supplementary Figs. 5-7d show the correlation between the correlation coefficients of paired neurons’ activities and their distance in each ensemble for individual fish.”

(Line 320-326): “These neurons tended to be localized at a closer distance with highly correlated activity (Fig. 4i; for individual fish, please see Supplementary Figs. 5-7d), although these ensembles did not have the particular brain region where the neurons within the ensembles of the same nature accumulated across different fish. Supplementary Figs. 9 and 10 show the distribution of the neurons within the two color rule-encoding ensembles, *i.e.* blue is dangerous and red is safe.”

(Line 453-459): “These neurons tended to be localized at a closer distance with highly correlated activity (Fig. 6j; for individual fish, please see Supplementary Figs. 5-7d, bottom right panel, scenery flow prediction error) although these ensembles did not have the particular

brain region where the neurons within the putative SFPE-encoding ensemble accumulated across different fish. Supplementary Fig. 12 shows the distribution of the neurons within the putative SFPE-encoding ensemble across different fish.”

2) The manuscript describes results from many fish, that have been in some cases sub-selected for having populations of neurons with certain interesting activity patterns. The selection of interesting activity patterns and fish appears to have been done by eye. That is fine, but also means that a substantial amount of subjectivity in the study. Is it possible to use a more principled or automated way of deciding which activity patterns to focus on? How is the problem of multiple comparisons addressed (that if you have enough fish or neurons, even if they are random, you will find some that look interesting)?

Response: In the manuscript, we described the ensembles which we focused on in the order of blue perception (new Fig. 3), blue is dangerous (new Fig. 4), red is safe (new Fig. 4) and SFPE (new Fig. 6). However, actually, we selected these ensembles first by observing the activities of NMF ensembles under the open loop condition as shown in two flow charts (new Supplementary Figs. 18 and 19). By following these two flow charts, we show how we selected the ensembles for blue perception, rule coding for blue and red color, and the scenery flow prediction error. The procedure is very simple and clear which may not need automatization. We newly added the following statement in Methods (Line 966-981).

“Identification of each ensemble

To identify the ensembles encoding blue perception (Fig. 3), blue is dangerous (Fig. 4), red is safe (Fig. 4) and SFPE (Fig. 6), we introduced two flow charts (Supplementary Figs. 18 and 19). In these flow charts, we first selected the candidates that encode the information we focused on by observing the activities of NMF ensembles under the open loop condition. Among the ensembles which showed increased activity under the open loop condition, the ensembles which met the criteria in the flow chart in Supplementary Fig. 18 and 19 were regarded as blue perception, the two color rules and putative SFPE-encoding ensembles. The ensembles which did not show the specificity in the activation to each color under the closed loop condition or the constant activation in GO or NOGO trials under the open-loop condition or only showed increased activity to red in GO or NOGO trials were abandoned. For the fish which did not experience the open-loop condition but rule reversal, we first observed the

activity of ensembles after establishment of behavioral learning instead of under the open loop condition and followed the second step in the two flow charts (Supplementary Figs. 18 and 19).”

Supplementary Fig. 18

Flow chart to identify the ensembles encoding blue perception, the rule that blue is dangerous and scenery flow prediction error (SFPE)

Flow chart to identify the ensembles encoding the rule that red is safe

However, to address the Reviewer 2's request on the problem of multiple comparison, we semi-automatized the first step of the selection procedure in the two flow charts by setting a threshold. After checking the usefulness of the semi-automatized selection method (new Supplementary Fig. 15 shows the result of Fish 1 in Supplementary Table1 as an example), we applied this semi-automatized selection method to the NMF ensembles generated by random shuffling of the original data to address the multiple comparison problem (new Supplementary Fig. 16 shows the result of Fish 1). We newly added the result in Discussion as follows (Line 567-574).

"We addressed the problem of multiple comparison to exclude the possibility that these ensembles were identified by chance. For this purpose, we created the semi-automatized selection method and applied it to the NMF ensembles generated from the neural population with randomly shuffled neural activity (for detail please see Methods and Supplementary Figs. 15 and 16). As a result, no ensemble was found to meet the criteria in the automatized selection method (Supplementary Fig. 16), suggesting that the ensembles we selected were not obtained by chance."

To show the detailed procedure above, we newly added the statement as follows in Methods (Line 983-1008).

“Multiple comparison problem

To address the problem of multiple comparison, we semi-automatized the procedure for selecting the ensembles of our interest. We first calculated the correlation coefficient between environment-dependent variables, $B(t)$ and $R(t)$, and NMF ensembles calculated from shuffled neural activity under the open loop condition in each fish. $B(t)$ is +1 if fish is in the

blue region, but otherwise the variable is 0. If the correlation coefficient to environment-dependent variables was higher than 0.25 (a threshold for the selection), the ensembles were regarded as candidates of the ensembles encoding the information we focused on. Among the candidates, the ensembles that met the criteria in the flow charts in Supplementary Fig. 18 and 19 were regarded as the ensemble encoding blue perception, the two color rule and putative SFPE-encoding ensembles. We first applied them to the original data from Fish1 in Supplementary Table1 as an example (Supplementary Fig. 15), showing the validity of these procedures for semi-automatic selection of the candidate ensembles of our interest. By further using this method, we confirmed that the correlation coefficients rarely reached the value higher than the threshold value 0.25 by calculating correlation coefficient between NMF ensembles made from randomly shuffled neural activities derived from the Fish1 original data and $B(t)$ or $R(t)$ (Supplementary Fig. 16). For the shuffling, two values in each neuron's activity in the time line were randomly selected and swapped, and this process was repeated n times, where n is the total bin number of the time line of each neuron's activity. We generated three sets of shuffled neural activity in each fish and these three shuffled neural activity sets were processed for NMF after determination of the number of typical ensembles by calculating AIC in each set. After NMF calculation, we calculated the correlation coefficient between the environment-dependent variables $B(t)$ and $R(t)$ and all NMF temporal time lines in each fish."

3) The maximization of reward model, and an internal model, are not necessarily incompatible or distinct, it just so happens that these have been formulated separately. The introduction is currently phrased as if these two models are distinct (using "in contrast" and "alternative model"). I suggest changing this to reflect that it's not necessarily one or the other (as, indeed, the authors themselves conclude based on the data).

Response: We changed introduction to indicate that two models are not distinct as follows (Line 19-21, 34-40).

(Line 19-21): "Animals make decisions under the principle of reward value maximization **and** surprise minimization. It is still unclear how these principles are represented in the brain and are reflected in behavior."

(Line 34-40): "One prevailing model underlying this behavioral process is based on the idea

that the ultimate aim of choice is to maximize utility or reward¹. **In addition to this**, adaptive behavior requires animals to generate the internal model of their environment and to take actions to minimize surprise (*i.e.* improbability) about the state they encounter in comparison with the state predicted from the internal model^{2,3}. **How these mechanisms are actually adopted by animals and are reflected in their behavior remains unknown^{4,5}.**

4) Line 49 "an example of the simplest behavioral paradigm", perhaps "simple" is better, since one can probably invent paradigms that are even simpler.

Response : We changed "simplest" to "simple". (Line 65)

5) Line 59, "Its brain is very small", it depends what one compares it to, perhaps quantifying it is better?

Response: We changed it with quantified data in the previous studies as follows (Line 54-55).

"The zebrafish brain is very small (3 mm^3)¹⁵ as compared to that of mice (509 mm^3)¹⁶ or humans (1400 cm^3)¹⁷ "

6) Line 66, comparison to mice and monkeys. That might be true, but maybe that's because of unnatural lab conditions or so. It should be avoided to make it seem that the authors are suggesting that adult zebrafish generally learn faster than mice and monkeys.

Response: We changed this statement to simply say "These tasks are commonly used also in mice and monkeys²²⁻²⁵". (Line 71-72)

7) The claim that only fish with the second ensemble encoding the state prediction error swim straight to the goal made me wonder, not the entire telencephalon is imaged, right? How confident are the authors that this population doesn't exist in the other fish, but was missed by the two-photon microscope?

To address this point, *i.e.* why we can argue that fish with SFPE ensemble swim straight to the goal under the situation that we could not image the entire telencephalon, we newly extended imaging into deeper telencephalic region by increasing the number of imaging planes from three to six (new Fig. 1c, red-boxed area and new Supplementary Fig. 14a which showed the average image of each plane in six-plane

calcium imaging). We newly described the results in the text as follows (Line 484-499). From the result, although we could not perform the whole pallium imaging, it is reasonable to think that the possibility that we missed the SFPE-encoding ensemble due to the low coverage rate might be small.

“ The putatively SFPE-encoding ensemble was observed in eight out of 24 fish that possessed a color rule-encoding ensemble. We examined the possibility that the failure to detect the putatively SFPE-encoding ensemble in the remaining 16 fish might result from the limitation in the scanning volume of the telencephalon. To address this question, we extended imaging into a deeper telencephalic region by increasing the number of imaging planes from three to six (Supplementary Fig. 14a shows the averaged calcium images of six planes). NMF calculations were performed in the deeper three planes and the superficial three planes separately and compared. In one out of four learner fish, we identified the putatively SFPE-encoding ensemble. Consistent with the previous results, this fish showed shorter halt periods than the fish with only the color rule-encoding ensemble (Supplementary Fig. 14b). Supplementary Fig. 14b shows the halt period of the two groups. The putatively SFPE-encoding ensemble was observed in the surface planes but not in the deeper planes (Supplementary Table 2 summarizes results of identified ensembles in surface and deeper planes in the four fish), implying that the surface three-plane imaging might be enough to capture the putatively SFPE-encoding ensemble. ”

Further, to exclude the possibility that the difference in the halt time between the two groups of fish was accidentally observed, we performed the data random shuffling as we newly described in Line 500-509 as the following.

“The permutation test of randomly dividing all 28 fish into two groups of 9 and 19 fish revealed that the difference of halt periods between two groups randomly made for 1000 times rarely exceeded the actually observed difference between the two groups categorized by the presence or absence of a putatively SFPE-encoding ensemble (permutation P -value= 0.011 < 0.05), confirming that the shorter halt period of fish with both color rule and putatively SFPE-encoding ensembles was not accidental.

Therefore, this statistical analysis confirmed that the fish with both ensembles swam forward more efficiently, *i.e.* swam straight toward the goal, than the fish with only the ensemble assigning a ‘blue-dangerous’ rule, which paused for a longer period on the way to the goal (Fig. 7a, b and Supplementary PowerPoint file). ”

8) The assemblies can be represented in space, as was done in the supplementary figures 4-6 and 8. It would be good to have this in the main text, also in 3d, and ideally also combined across fish, so readers can see if there are any spatial patterns where certain areas of the brain preferentially have certain neuron types.

Response: Actually, we could not identify the brain area in which neurons in each ensemble are preferentially distributed. However, we newly found that neurons within the same ensemble show tendency that closer neurons are more correlated in their time-lapse activity changes (new Figs. 3g, 4i, 6j). Therefore, we mentioned about this in the main text (For blue perception, Line 265-275; for color rules, Line 320-326; for SFPE, Line 453-459) and put the data in the main figure, instead of putting the spatial distribution in the main figures. However, as Reviewer 2 suggested, we added the spatial distribution of neurons which contribute to the ensembles encoding blue perception, blue is danger, red is safe and SFPE in all fish in Supplementary Figs. 8, 9, 10, 12, respectively.

(Line 265-275): “However, to further evaluate the spatial distribution of the neurons within the ensemble, we calculated the correlation coefficient of the ten neurons with the highest levels of contribution to the ensemble. We then plotted the relationship between the paired correlation coefficients and the distance of these neurons after averaging the data derived from all 27 fish that had the ensemble encoding blue color perception (Fig. 3g; for individual fish, please see Supplementary Figs. 5-7d, upper left panels, blue perception). The result showed that neurons with highly correlated activity have a tendency to be localized at a closer distance, implying that neurons in the zebrafish telencephalon encode the information in locally clustered populations. Supplementary Figs. 5-7d show the correlation between the correlation coefficients of paired neurons’ activities and their distance in each ensemble for individual fish.”

(Line 320-326): “These neurons tended to be localized at a closer distance with highly correlated activity (Fig. 4i; for individual fish, please see Supplementary Figs. 5-7d), although these ensembles did not have the particular brain region where the neurons within the ensembles of the same nature accumulated across different fish. Supplementary Figs. 9 and 10 show the distribution of the neurons within the two color rule-encoding ensembles, *i.e.* blue is dangerous and red is safe.”

(Line 453-459): “These neurons tended to be localized at a closer distance with highly correlated activity (Fig. 6j; for individual fish, please see Supplementary Figs. 5-7d, bottom

right panel, scenery flow prediction error) although these ensembles did not have the particular brain region where the neurons within the putative SFPE-encoding ensemble accumulated across different fish. Supplementary Fig. 12 shows the distribution of the neurons within the putative SFPE-encoding ensemble across different fish.”

9) Line 83, "reiterating" is not the right word here, perhaps "repeating" or "iterating" is better.

Response: We removed this statement. Therefore, the correction is not necessary.

10) What is the motivation for using *camk2a* and *vglut2a* promoters?

Response: To increase the number of the observed excitatory neurons in the dorsal pallium region, we used *camk2a* and *vglut2a* promoters (Cover letter Fig. 1). We put this figure in this cover letter but not in the figures in the manuscript. If the editor and Reviewer 2 think it should be included in the figures in the manuscript, we will put them in the manuscript.

Cover letter Fig. 1

TgBAC(camk2a:GAL4VP16)^{rw0154a}; TgBAC(vglut2a:Gal4); Tg(UAS:G-CaMP7)^{rw0155}

TgBAC(camk2a:GAL4VP16)^{rw0154a}; Tg(UAS:G-CaMP7)^{rw0155}

11) The description of figure 2g is confusing to me. Why is there "after reaching goal" only shown in the "nogo template" and not in the "go template"? And are these for only the six selected fish or all of them? How are the fish selected, by eye or by some automated criterion?

Response: Although we put the data from all fish of all cases in previous and new Supplementary Fig. 2, we put the "after reaching goal in GO trials of GO template" data into new Fig. 2g. The data presented in new Fig.2g is the data from one fish but the data from all fish are presented in new Supplementary Fig. 2. Therefore, it should be enough for the reader to know the results of template matching analysis.

12) Lines 209-214, it seems important to stress that this was not the case before learning.

Response: We changed the text to "The similarity increase was not observed in the adaptation and initial stages of training." (line 212-213)

13) Line 229, what does "definite" mean here, and is it the right word?

Response: We would like to indicate the existence of some ensemble responding to blue, red color perception. Therefore, we change "definite" to "some". (Line 231)

14) Figure 3D is not referred to I think.

Response: We cited new Fig. 3d in the text. (Line 237)

15) The figures can be improved. I found them confusing because a lot of labels and legends are lacking, and one has to hunt in the figure legend for them. For example, in 2e, the blue and red labels "Index to..." refer to the blue and red traces, and not to the blue and red horizontal lines, which represent the stimulus. That is really confusing: please also label the blue and red horizontal lines, and explain what the higher and the lower position means (go and nogo). Please label the tail angle, the black displacement line, and the red line. What is the horizontal black line under the yellow trace?

Response: We followed Reviewer 2's comment and improved new Fig. 2e as the following.

We added the labels for GO, NOGO period, tail bend angle, traveled distance (the black displacement line) and goal border (the red line) to the enlarged Fig. 2e. The horizontal black line under the yellow trace in the previous figures was only the border of each graph which is not data. To avoid confusion, we removed the horizontal black line.

16) Similarly it is not so clear what each dot represents for example in 2g. Is one dot a fish, or a trial? Please clarify that.

Response: The dots in Fig.2g indicate the trials. To make it clear, we added "circles indicate the peaks in value in each trial". (Line 192-193)

17) Line 980, n=32 trails or fish? Please specify both.

Response: To make the meaning of the number clear, we removed the representation of "n= X" and instead of that we used "X fish" or "X trials" in the whole manuscript. (Line 156, 157, 1262-1263)

18) Line 1000, "color of the apposition of the fish", I think that can be easier understood if it's something like "color of the environment at the current position of the fish".

Response: We changed “color of the apposition of the fish” to “color of the environment at the position of the fish” (Line 1282-1283 and 1313)

19) Figure 3c and after, here the yellow trace is labeled but it should be "bend" instead of "bent".

Response: We changed the word “bent” to “bend” (new Figs. 2e, 3b-c, 4-6, line1283, 1314).

20) Several grammar errors need to be fixed, I can help with this in the next round.

Response:

Although we tried our best to prevent errors, and received proof corrections by a native editor, **we would greatly appreciate if we could get further help from you.**

REVIEWER COMMENTS

Reviewer #1 (Remarks to the Author):

In their response to reviewer concerns raised Torigoe et al. substantially improved the manuscript and their treatment of the data. As expected not all concerns could be fully mitigated, however, since the authors toned down the language, I think that the claims of the paper are now supported well enough by the data to warrant publication. In particular my major concern, that "there are major concerns with the current implementation of the task and the conclusions drawn" has been sufficiently addressed through new analyses, toning down of the conclusions and clarifications of the behavioral task itself.

There is still a slight worry related to coverage of the telencephalon as evidenced by the lack of the "blue encoding" ensemble even in some fish with the six-plane imaging. However, the authors present a convincing enough case that most ensembles should be confined to the imaged dorsal part. As I stated previously, I think that the overall approach and use of the zebrafish model in general hold great promise to understand the neural basis for learned behavior, and therefore, together with the present improvements I do think that this paper should be published in Nature Communications.

Reviewer #2 (Remarks to the Author):

In the revised manuscript, the authors took great care to address all the reviewer's comments. The new version is much improved. I have a few remaining minor comments:

Following the discussion of non-negative matrix factorization and neural ensembles, I feel the comment is not yet resolved because it is still unclear to me how a NMF component is turned into an ensemble. In a component, every neuron receives a weight, which can be zero or positive. The concept of a neural ensemble suggests that each ensemble consists of a (small) subset of the total number of neurons. But in NMF it is quite common that many neurons get some weight above zero, although it can be small. Was an ensemble defined as all neurons that had a weight greater than zero? Or was an ensemble defined as all neurons having a weight above some threshold, and if so, how was the threshold determined? Whichever method was chosen, what was the statistics of ensemble size relative to the total number of neurons imaged?

Next, regarding the question of why *camk2a* and *vglut2a* were used as promoters for the imaged animals. My question was mostly about why not use a pan-neuronal promoter and image all neurons in those areas. Was there a special reason to focus on these particular labeled populations?

Finally, I am happy to check the grammar at final stage of the manuscript, although it is already good and I did not find issues. At the moment I have little time and don't want to delay this review, but please feel free to send it back to me for a final pass.

Point-by-point responses to the comments

Reviewer #1 (Remarks to the Author):

In their response to reviewer concerns raised Torigoe et al. substantially improved the manuscript and their treatment of the data. As expected not all concerns could be fully mitigated, however, since the authors toned down the language, I think that the claims of the paper are now supported well enough by the data to warrant publication. In particular my major concern, that “there are major concerns with the current implementation of the task and the conclusions drawn” has been sufficiently addressed through new analyses, toning down of the conclusions and clarifications of the behavioral task itself.

There is still a slight worry related to coverage of the telencephalon as evidenced by the lack of the “blue encoding” ensemble even in some fish with the six-plane imaging. However, the authors present a convincing enough case that most ensembles should be confined to the imaged dorsal part. As I stated previously, I think that the overall approach and use of the zebrafish model in general hold great promise to understand the neural basis for learned behavior, and therefore, together with the present improvements I do think that this paper should be published in Nature Communications.

Response: Thank you for your comments. Thanks to your important comments, our manuscript was dramatically improved.

Reviewer #2 (Remarks to the Author):

In the revised manuscript, the authors took great care to address all the reviewer's comments. The new version is much improved. I have a few remaining minor comments:

Following the discussion of non-negative matrix factorization and neural ensembles, I feel the comment is not yet resolved because it is still unclear to me how a NMF component is turned into an ensemble. In a component, every neuron receives a weight, which can be zero or positive. The concept of a neural

ensemble suggests that each ensemble consists of a (small) subset of the total number of neurons. But in NMF it is quite common that many neurons get some weight above zero, although it can be small. Was an ensemble defined as all neurons that had a weight greater than zero? Or was an ensemble defined as all neurons having a weight above some threshold, and if so, how was the threshold determined? Whichever method was chosen, what was the statistics of ensemble size relative to the total number of neurons imaged?

Response: To clarify that only a part of the imaged neurons actively contributed to the ensemble, we newly calculated the ratio of the number of neurons which had a contribution value larger than zero in each ensemble to the number of total imaged neurons in each seven fish which possessed all ensembles mentioned in this manuscript (Fish1-7 in Supplementary Table 1) and added the results as new Supplementary Fig. 8. This suggests that the neurons which have a contribution larger than zero in each ensemble are a subset of total imaged neurons.

We also simply mention about this in the text as the followings,

(Line 243-244, page 9) for the blue-perception ensemble, “In this ensemble, approximately 20% of the total imaged neurons showed larger than zero contribution within the ensemble (Supplementary Fig 8).”

(Line 316-318, page12) for the blue-is-dangerous and red-is-safe ensembles, “In these blue and red responsive ensembles, approximately 20% of the total imaged neurons showed larger than zero contribution within the ensemble (Supplementary Fig 8).”

(Line 451-452, page 16) for the scenery-flow-prediction error ensemble, “In this ensemble, approximately 20% of the total imaged neurons showed larger than zero contribution within the ensemble (Supplementary Fig 8).”

Next, regarding the question of why *camk2a* and *vglut2a* were used as promoters for the imaged animals. My question was mostly about why not use a pan-neuronal promoter and image all neurons in those areas. Was there a special reason to focus on these particular labeled populations?

Response: We agree that the best way to reveal the role of the dorsal pallium is simultaneous imaging of excitatory and inhibitory neurons in the dorsal pallium. However, the only currently available promoters for the pan-neuronal expression are those of the *elavl3* or *neuroD* genes. Since these genes are activated only immediately after neural differentiation, they have been used mainly to label neurons in the larval zebrafish, and it is not precisely known how much and variably the genes under the control of these promoters are expressed in different populations of neurons in the adult brain. Therefore, in this study, as a first step, we decided to focus on the excitatory neurons. Imaging and analyzing the activity of inhibitory neurons by using specific promoters is currently ongoing in our group. We would like to report the results of this study in the future to generate a more integrated view of the entire neural activity.

Finally, I am happy to check the grammar at final stage of the manuscript, although it is already good and I did not find issues. At the moment I have little time and don't want to delay this review, but please feel free to send it back to me for a final pass.

Response: Thank you for your kindness.

REVIEWERS' COMMENTS

Reviewer #2 (Remarks to the Author):

All my comments have been addressed. Thank you for clarifying the NMF results. The paper looks very good. Congratulations on a very nice study.

Point-by-point responses to the comments

REVIEWER COMMENTS

Reviewer #2 (Remarks to the Author):

All my comments have been addressed. Thank you for clarifying the NMF results.
The paper looks very good. Congratulations on a very nice study.

Response: Thank you for your comments. Thanks to your important comments, our manuscript was dramatically improved.